# Adaptive Conformal Prediction via Mixture-of-Experts Gating Similarity

**Jingsen Kong**[*1,2], **Wenlu Tang**[*2], **Dezheng Kong**[2], **Linglong Kong**[2], **Guangren Yang**[1†]**& Bei Jiang**[2†]

[1]Jinan University, Guangzhou, China
[2]University of Alberta, Edmonton, Canada
{jingsen0111@stu2023.jnu,tygr@jnu}.edu.cn
{wenlu4,dezheng2,bei1,lkong}@ualberta.ca

## Abstract

Prediction intervals are essential for applying machine learning models in real applications, yet most conformal prediction (CP) methods provide coverage guarantees that overlook the heterogeneity and domain knowledge that characterize modern multimodal datasets. We introduce Mixture-of-Experts Conformal Prediction (MoE-CP), a flexible and scalable framework that uses the gating probability vectors of Mixture-of-Experts (MoE) models as soft domain assignments to guide similarity-weighted conformal calibration. MoE-CP weights calibration residuals according to the similarity between gating vectors of calibration and test points, producing prediction intervals that adapt to latent subpopulations without requiring explicit domain labels. We provide theoretical justification showing that MoE-CP preserves nominal marginal validity under common similarity measures and improves conditional adaptivity when the gating captures domain structure. Empirical results on synthetic and real-world datasets demonstrate that MoE-CP yields more domain-aware, interpretable, and often tighter intervals than existing conformal baselines while maintaining target coverage. MoE-CP offers a practical route to reliable uncertainty quantification in latent heterogeneous, multi-domain environments.

## 1 Introduction

Modern machine learning increasingly handles heterogeneous and latent multi-domain data. By heterogeneous, we mean data whose characteristics such as noise level, feature distribution, or label variability that differ across regions of the input space. By multi-domain, we mean that these differences often align with implicit or explicit groups (Hahnloser & Seung, 2000), such as patients from different hospitals, users across geographic regions, or sensor modalities in autonomous systems. Specifically, such domains are not always labeled in advance, and they emerge as latent sub-populations embedded in the data. In these settings, a single "one-size-fits-all" predictive uncertainty such as standard conformal prediction (Vovk et al., 2005; Lei et al., 2018) is not adequate. Intervals that are valid on average may be overly wide in low-variance domains and narrow in high-variance ones. Reliable prediction intervals must therefore adapt to the underlying domain structure in both observed or latent patterns while preserving rigorous coverage guarantees.

To address this challenge, we introduce Mixture-of-Experts Conformal Prediction (MoE-CP), a new method that combines conformal prediction with the representational power of Mixture-of-Experts (MoE) models (Jacobs et al., 1991). MoE architectures decompose complex prediction tasks into a set of specialized expert functions, each responsible for modeling a sub-regime of the data. A gating network assigns a probability vector over experts for each input, effectively acting as a soft domain classifier. This structure has proven highly successful in scaling large language models (Shazeer et al., 2017), multimodal learning (Nguyen et al., 2023), and domain adaptation, precisely because it can capture heterogeneous or multi-domain structure without explicit domain labels. In our setting, the gating vector plays a central role that it provides a natural similarity measure that identifies which

---

*Equal contributions in alphabetical order
†Corresponding author

calibration points belong to the same latent regime as a test point. By weighting calibration residuals according to this similarity, MoE-CP tailors prediction intervals to the local predictive regime of the test input. In effect, the MoE component solves the challenge of latent domain identification, while the conformal component ensures rigorous coverage guarantees. Together, this integration yields intervals that are both valid and adaptively sensitive to heterogeneous, multi-domain data. We show the workflow of our framework in Figure 1.

This modification fundamentally adapts conformal prediction to data complexity. In heterogeneous or multi-domain datasets, test points may lie in regions with very different conditional variances or expert specialization. MoE-CP adjusts intervals using the most relevant calibration points, becoming wider in noisy regions and narrower where predictions are reliable. In multi-domain data, the gating distribution naturally links different domains, so intervals adapt across domains without manual splitting. This work is related to randomly localized conformal prediction (RLCP) (Hore & Barber, 2025), where calibration points are chosen according to their proximity in the covariate space. By contrast, MoE-CP identifies relevant calibration points based on proximity in a *learned, label-aware latent regime space* defined by the gating probabilities of the mixture-of-experts. This makes the intervals not only valid, but also sharper, more flexible, and easier to interpret than other local weighted conformal methods, see Section 5 and Remark 5.

In this work, we also provide a theoretical analysis showing that MoE-CP (i) preserves marginal validity when using KL or cross-entropy divergences, (ii) retains asymptotic validity under a wide class of similarity measures, and (iii) approaches conditional coverage under a mixture representation under some conditions. Empirically, we show that MoE-CP consistently improves efficiency and adaptivity across synthetic experiments and real multidomain data.

In summary, we highlight our contributions as follows.

- We propose MoE-CP, a conformal prediction framework that leverages MoE gating vectors to adapt intervals to local domain structure.

- We establish theoretical guarantees for coverage under KL, cross-entropy, and general divergences, as well as approximate conditional coverage under a mixture representation.

- We demonstrate empirically that MoE-CP yields domain-adaptive intervals compared to existing conformal methods, particularly in heterogeneous and multi-domain settings.

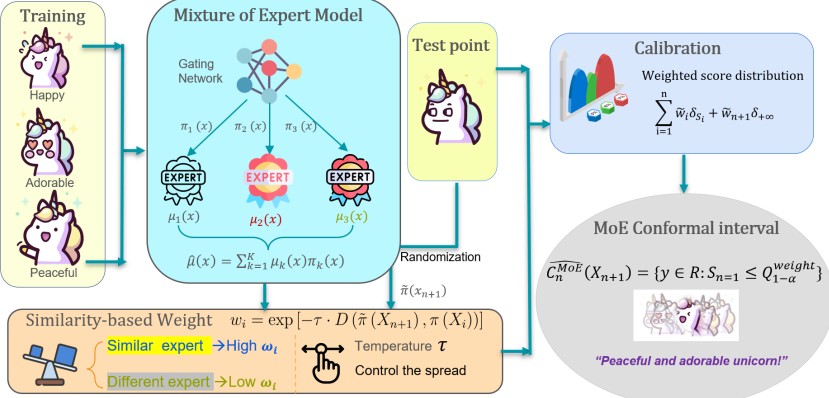

Figure 1: A MoE model assigns each input to experts through a gating network, producing both predictions and soft domain probabilities. Calibration residuals are weighted by the similarity of gating vectors. The final conformal interval adapts automatically across domains.

## 2 METHODOLOGY

### 2.1 PROBLEM SETUP AND NOTATIONS

We work in a supervised prediction setting where the goal is to produce valid prediction intervals for unseen data, potentially coming from multiple latent domains. Let

$$\mathcal{D}_{\text{train}} = \{(X_i, Y_i)\}_{i=1}^m, \quad \mathcal{D}_{\text{cal}} = \{(X_i, Y_i)\}_{i=1}^n$$

be the training and calibration set respectively. Both are drawn independently and identically distributed (i.i.d.) from an arbitrary distribution $P$ that contains $K$ latent domains. A prediction set $\hat{\mathcal{C}}_n(X_{n+1})$ is called *marginally valid* interval if for any new test point $(X_{n+1}, Y_{n+1}) \sim P$,

$$\mathbb{P}\{Y_{n+1} \in \hat{\mathcal{C}}_n(X_{n+1})\} \geq 1 - \alpha,$$

where $\alpha \in (0, 1)$ is the desired miscoverage rate, and the probability $\mathbb{P}$ is over the randomness in training, calibration, and test sampling. While marginal validity is a standard target in conformal prediction, it does not guarantee coverage for each specific covariate $x$. The stronger property is *conditional validity*, that is, for any $x$,

$$\mathbb{P}\{Y_{n+1} \in \hat{\mathcal{C}}_n(X_{n+1}) \mid X_{n+1} = x\} \geq 1 - \alpha, \quad \forall x.$$

In practice, exact conditional validity is unattainable without strong assumptions (Lei & Wasserman, 2014; Barber et al., 2021), but it motivates our method: assigning higher influence to calibration points that are most similar to the test point.

### 2.2 MIXTURE-OF-EXPERTS (MoE) PREDICTOR

We consider the observed data are input-output pairs $(X_i, Y_i) \in \mathcal{X} \times \mathbb{R}$, which in practice may arise from latent and unknown multi-domain structure. However, these domain assignments are not observed. What we see $(X_i, Y_i)$ is drawn from an overall distribution $P$, which we treat as exchangeable. Thus, although the data-generating mechanism is inherently heterogeneous and multi-domain, the observed dataset appears as a single mixed population.

To model this heterogeneous and multidomain data, we use a *Mixture-of-Experts* (MoE) architecture. The MoE partitions the input space into $K$ sub-regimes and assigns each to a specialized expert function $\mu_k(X) : \mathcal{X} \to \mathbb{R}$. Given an input $X$, a gating network produces probability vector

$$\pi(X) = (\pi_1(X), \ldots, \pi_K(X)) \in [0, 1]^K, \quad \sum_{k=1}^K \pi_k(X) = 1,$$

where each $\pi_k(X)$ that approximates the latent domain probabilities $\pi_k^*(X)$ which reflects the degree to which expert $k$ is responsible for predicting $X$. Each $\pi_k(X)$ is parameterized via a softmax:

$$\pi_k(X) = \frac{\exp(\ell_k(X))}{\sum_{j=1}^K \exp(\ell_j(X))}, \tag{1}$$

where $\ell_k(X)$ is a logit score for expert $k$ of given $X$, from a Multi-layer perceptron (MLP) or transformer block. The expert functions $\mu_k$ and gating logits $\ell_k$ are trained jointly on the training dataset $\mathcal{D}_{\text{train}}$ to minimize a loss function such as mean squared error. The choice of $K$ is discussed in Section A.3. The MoE prediction is then defined as the weighted average of each expert:

$$\hat{\mu}(X) = \sum_{k=1}^K \pi_k(X)\mu_k(X), \tag{2}$$

where $\mu_k(X)$ denotes the output of the $k$-th expert model, and $\pi_k(X) \in [0, 1]$ is the gating probability assigned to expert $k$ at input $X$ defined by (1). After training, each prediction $\hat{\mu}(X)$ comes with an interpretable domain probability vector $\pi(x)$. This allows us to exploit domain structure in a data-driven way when constructing adaptive conformal prediction intervals.

## 2.3 MoE-based Similarity Weighted Conformal

We now describe how to construct valid and adaptive prediction intervals using the domain structure captured by the MoE model. Our approach extends standard conformal prediction by assigning higher importance to calibration points that share similar domain affinities with the test point, where similarity is measured in the MoE's learned gating space.

Given the trained MoE model from Section 2.2, we first define a nonconformity score $S$ for each calibration point. While conformal prediction can be applied with many different scores, in this work we use the absolute residual:

$$S_i = |Y_i - \hat{\mu}(X_i)|, \quad (X_i, Y_i) \in \mathcal{D}_{\text{cal}},$$

where $\hat{\mu}(X_i) = \sum_{k=1}^{K} \pi_k(X)\mu_k(X)$ is the MoE's point prediction for $X_i$ and $\pi_k(X_i)$ represents the model's gating probability that input $X_i$ belongs to expert/domain $k$. The MoE also outputs a vector $\pi(X_i) = (\pi_1(X_i), \ldots, \pi_K(X_i))$.

While we focus on absolute residual-based scores (Lei et al., 2018) for clarity, our MoE-weighted framework is general and easily extensible to other conformity scores, such as quantile (Romano et al., 2019) or distributional conformity (Chernozhukov et al., 2021a) scores. These extensions would allow MoE-CP to adapt not only to mean prediction errors but also to richer forms of predictive uncertainty, further enhancing its applicability in heteroscedastic or multi-domain environments.

**Remark 1** *In heterogeneous or multidomain datasets, calibration points are not equally informative for every test point. Two points with the same covariates $X$ distribution but different domain label may exhibit different residual distributions. The gating vector $\pi(x)$ acts as a soft domain assignment, capturing the model's internal partitioning of the input space. Thus, calibration points with similar $\pi(\cdot)$ to the test point are statistically closer in the model's latent space, and their residuals provide a more accurate basis for interval calibration.*

**Randomized gating vector for validity.** For a test input $X_{n+1}$, we first obtain its gating vector $\pi(X_{n+1})$. To ensure coverage validity, we introduce randomization following the principles of Zhang & Candès (2024) and Hore & Barber (2025). Specifically, we generate:

$$\tilde{L} \sim \text{Multinomial}(\tau, \pi(X_{n+1})), \quad \tilde{\pi}(X_{n+1}) = \frac{\tilde{L}}{\tau}, \tag{3}$$

where $\tau \in \mathbb{N}^+$ is a temperature parameter controlling the randomization magnitude such that small $\tau$ introduces more randomization and involves greater variability in weights, while large $\tau$ will let $\tilde{\pi}$ approach $\pi(X_{n+1})$ and introduce less variability. This randomized $\tilde{\pi}(X_{n+1})$ plays the same role as randomized covariates in localized conformal prediction (Hore & Barber, 2025) but is computed in gating space rather than the raw input space.

**Similarity-based weighting.** Our key idea of the weighting lies on the fact that calibration points with similar gating vectors $\pi(x)$ to the test point should have more influence. And thus we formalize similarity via a divergence between probability vectors. Given $\tilde{\pi}(X_{n+1})$ and each gating vector $\pi(X_i)$, we compute similarity-based weights:

$$w_i = \exp\left[-\tau \cdot D\left(\tilde{\pi}(X_{n+1}), \pi(X_i)\right)\right], \tag{4}$$

where $D(\cdot, \cdot)$ is a divergence between probability vectors. The weights are then normalized across calibration points $\mathcal{D}_{\text{cal}} = \{(X_i, Y_i), i \in \mathcal{I}_{\text{cal}} = \{1, \cdots, n\}\}$ and the test point:

$$\tilde{w}_i = \frac{w_i}{\sum_{j=1}^{n+1} w_j}, \quad i \in \mathcal{I}_{\text{cal}} \cup \{n+1\}. \tag{5}$$

This normalization ensures that weights form a probability distribution over the calibration set plus the testing point (Tibshirani et al., 2019; Barber et al., 2023).

**Remark 2** *Common choices of $D(\cdot, \cdot)$ include Kullback-Leibler divergence, cross-entropy, cosine distance, or Euclidean distance. It aims to measure the similarity between $\tilde{\pi}(X_{n+1})$ and $\pi(X_i)$. Larger similarity between $\tilde{\pi}(X_{n+1})$ and $\pi(X_i)$ results in smaller divergence, and thus carry out higher weight $w_i$.*

**Remark 3** *Although there is no well-established existing works on choice of $\tau$, we show some practical implications to guide our choice. If $\pi(x)$ is accurate and stable, we prefer larger $\tau$ to exploit domain-specific residuals for tighter intervals. However, if $\pi(x)$ is noisy or high-variance, smaller $\tau$ will be chosen to avoid over-committing to potentially spurious domain assignments. As extreme cases, setting $\tau \to 0$ reduces the method to standard split conformal prediction with uniform weights, while letting $\tau \to \infty$ places essentially all weight on the test point itself.*

**MoE-weighted conformal interval.** The final MoE-weighted conformal prediction interval is:

$$\hat{\mathcal{C}}_n^{\mathrm{MoE}}(X_{n+1}) = \left\{ y \in \mathbb{R} : |y - \hat{\mu}(X_{n+1})| \le Q_{1-\alpha}\left( \sum_{i=1}^{n} \tilde{w}_i \delta_{S_i} + \tilde{w}_{n+1} \delta_{+\infty} \right) \right\},$$

where $Q_{1-\alpha}(\cdot)$ is the weighted empirical ($1-\alpha$)-quantile of the nonconformity scores. The term $\delta_{+\infty}$ corresponds to the standard conformal adjustment ensuring coverage (Tibshirani et al., 2019; Barber et al., 2023). This approach yields adaptive, valid, and interpretable prediction intervals that reflect the multimodal similarity structure inherent in the gating mechanism of MoE. It is particularly effective when training examples are heterogeneously distributed across latent regimes defined by multimodal signals. Algorithm 1 is the workflow of our MoE-weighted conformal framework, and the full pipeline of the framework is provided in Appendix A.1.

---

**Algorithm 1** MoE-Weighted Conformal Prediction

---

**Require:** $\mathcal{D}_{\mathrm{train}}, \mathcal{D}_{\mathrm{cal}}, X_{n+1}, S, K, D, \tau, \alpha$
**Ensure:** $\hat{\mathcal{C}}_n^{\mathrm{MoE}}(X_{n+1})$
1: **Fit MoE**: learn experts and gate $\{\mu_k, \pi_k\}_{k=1}^K$ on $\mathcal{D}_{\mathrm{train}}$ by (1); let $\hat{\mu}(X) = \sum_k \pi_k(X)\mu_k(X)$.
2: **Calibration**: for $i \in \mathcal{I}_{\mathrm{cal}}$, compute $s_i = S(X_i, Y_i)$ and $\boldsymbol{\pi}_i = \boldsymbol{\pi}(X_i) = (\pi_1(X_i), \ldots, \pi_k(X_i))$.
3: **Randomize**: compute $\pi_{n+1} = \pi(X_{n+1})$; draw $\tilde{L}$ and $\tilde{\pi}$ by (3).
4: **MoE weights**: For $i \in \mathcal{I}_{\mathrm{cal}} \cup \{n+1\}$, let $w_i = \exp\{-\tau D(\tilde{\pi}, \pi_i)\}$; normalize $\tilde{w}_i$ by (5).
5: **Quantile**: form $F = \sum_{i \in \mathcal{I}_{\mathrm{cal}}} \tilde{w}_i \delta_{s_i} + \tilde{w}_{n+1} \delta_{+\infty}$ and let $Q_{1-\alpha}$ be its $(1-\alpha)$-quantile.
6: **return** $\hat{\mathcal{C}}_n^{\mathrm{MoE}}(X_{n+1}) = \{y : S(X_{n+1}, y) \le Q_{1-\alpha}\}$.

---

**Remark 4** *The MoE-CP framework is flexible: users may specify the divergence measure, the number of experts $K$, and the temperature $\tau$. Our theoretical and empirical results show that MoE-CP is largely insensitive to these choices and admit simple practical defaults (see Appendix A.3 for guidelines and Appendix A.4.3 for empirical ablation results). In practice, we recommend using the KL divergence by default (it additionally satisfies exact marginal validity in our analysis); choosing a small $K$ (e.g., $K = 2, 3$) when the training/calibration sample size is modest (e.g., $n \le 1500$) and allowing larger $K$ when large amount of data is available; and setting $\tau$ to a moderate value (e.g., $\tau \in [100, 300]$) to balance adaptivity and effective sample size. When needed, standard model-selection procedures such as cross-validation or an over-parameterized gating network with sparsity regularization can be used to select or automatically prune $K$.*

**Remark 5** *The core idea of our construction is to assign larger weights to calibration points whose gating distributions are close to $\tilde{\pi}(X_{n+1})$, an estimate of the true gating probability $\pi(X_{n+1})$. A randomization step is included to guarantee marginal validity (see the next section for theoretical details). The novelty of our framework is to define conformal weights via MoE gating probabilities, which—by leveraging the strong predictive performance of MoE models—yields improved adaptivity and efficiency. Related localized conformal methods (Guan, 2023; Hore & Barber, 2025) weight calibration points by the Euclidean distance between covariates; such an approach may fail to identify latent domains or heterogeneity, and may perform poorly in high-dimensional settings where many features are irrelevant. Zhang & Candès (2024) instead fit a mixture model on the conformity scores to obtain weights, but this requires refitting the mixture for every candidate $Y_{n+1} = y$, which is computationally expensive. By contrast, our weights are computed from an MoE gating network trained once on the training data, making the procedure both adaptive and computationally efficient.*

**Remark 6** *The MoE-CP admits several natural extensions. It can be paired with alternative conformity scores; for example, one may use the conformalized quantile score (Romano et al., 2019; Sesia & Romano, 2021): $S(X, Y) = \max\{\hat{q}_{\alpha/2}(X) - Y, Y - \hat{q}_{1-\alpha/2}(X)\}$, where $\hat{q}_{\alpha/2}(\cdot)$ and $\hat{q}_{1-\alpha/2}(\cdot)$*

*are lower and upper quantile regressors jointly estimated within an MoE model. The framework also extends naturally to online and time-series settings (Gibbs & Candes, 2021); for instance, one can combine MoE-CP with time-aware MoE architectures (e.g., Time-MoE (Shi et al., 2024)) to produce adaptive, sequentially valid prediction intervals. Further details are given in Appendix A.6.*

## 3 THEORETICAL RESULTS

We now present theoretical guarantees for the proposed MoE-weighted conformal prediction. Our goal is to show that the method preserves valid coverage while adapting to domain similarity as reflected in the gating space.

**Theorem 1** *Suppose that $(X_i, Y_i), i \in [n+1]$, are exchangeable. For the KL-based or cross-entropy based divergence, and for any $\tau \in \mathbb{N}^+$, the MoE weighted conformal interval satisfies*

$$\mathbb{P}\{Y_{n+1} \in \hat{\mathcal{C}}_n^{\mathrm{MoE}}(X_{n+1}) \mid \tilde{\pi}(X_{n+1})\} \geq 1 - \alpha.$$

*Marginalizing over the randomization $\tilde{\pi}(X_{n+1})$, the MoE-weighted conformal interval is therefore marginally valid:*

$$\mathbb{P}\{Y_{n+1} \in \hat{\mathcal{C}}_n^{\mathrm{MoE}}(X_{n+1})\} \geq 1 - \alpha.$$

Theorem 1 shows that when the divergence used in weighting is KL divergence or cross-entropy, the constructed interval has the standard conformal property that covers the label of a new test point with probability at least $1 - \alpha$. Importantly, this holds conditional on the randomized gating vector $\tilde{\pi}(X_{n+1})$, and remains valid after marginalization. This result ensures that our method retains the baseline of conformal prediction even after introducing MoE-based similarity weighting. We next consider divergences other than KL. To formalize this, we impose the following mild assumption.

**Assumption 1** *Denote $\tilde{w}_{i,KL}$ as the KL-based weight, as $\tau \to \infty$, the similarity measure $D(\cdot, \cdot)$ satisfies $\tilde{w}_{i,D} = \tilde{w}_{i,KL} + o(1)$.*

Assumption 1 holds for any divergence measure that shares the same unique minimizer as the KL divergence; this claim is formalized in Proposition 1 in Appendix A.2. In particular, provided a sufficiently large calibration set, the assumption is satisfied by many divergences, e.g., Jeffreys divergence, Hellinger distance, Euclidean distance, etc. See Proposition 1 for a detailed discussion.

**Theorem 2** *Under the assumption in Theorem 1, for any similarity measure $D(\cdot, \cdot)$ satisfying Assumption 1, and assuming the nonconformity score $S$ has a continuous distribution, the MoE conformal interval based on $D(\cdot, \cdot)$ satisfies*

$$\mathbb{P}\{Y_{n+1} \in \hat{\mathcal{C}}_n^{\mathrm{MoE}}(X_{n+1}) \mid \tilde{\pi}(X_{n+1})\} \geq 1 - \alpha + o(1).$$

Theorem 2 states that even if KL divergence is replaced with other similarity measures, the resulting intervals still maintain the conformal coverage property up to a vanishing error term. In other words, validity is robust to the choice of divergence, while the choice of $D$ may influence efficiency. Finally, we analyze the case where the nonconformity scores have a latent finite mixture representation.

**Theorem 3** *Under the assumption in Theorem 1, suppose that the nonconformity score satisfies $S_i | X_i \sim \sum_{k=1}^{K} \pi_k^*(X_i) f_k^*$, where $\pi_k(X_i)$ is the true domain probability and $f_k^*$ is the score distribution within domain $k$. Define the **oracle** MoE-weighted conformal interval using the true gating probabilities:*

$$\hat{\mathcal{C}}_n^{\mathrm{MoE},*}(X_{n+1}) = \left\{ y \in \mathbb{R} : S(X_{n+1}, y) = Q_{1-\alpha}\left(\sum_{i=1}^{n} \tilde{w}_i^* \delta_{S_i} + \tilde{w}_{n+1}^* \delta_{+\infty}\right) \right\},$$

*where the oracle weight $\tilde{w}_i^*$ is defined as*

$$\tilde{w}_i^* = \frac{\exp[-\tau D(\pi^*(X_{n+1}), \pi^*(X_i))]}{\sum_{i=1}^{n+1} \exp[-\tau D(\pi^*(X_{n+1}), \pi^*(X_i))]}, \quad i \in [n+1],$$

*for any similarity measure $D(\cdot, \cdot)$ and $\tau \in \mathbb{N}^+$. Then,*

$$\mathbb{P}\{Y_{n+1} \in \hat{\mathcal{C}}_n^{\mathrm{MoE},*}(X_{n+1}) \mid X_{n+1}\} = \mathbb{P}\{Y_{n+1} \in \hat{\mathcal{C}}_n^{\mathrm{MoE},*}(X_{n+1}) \mid \pi^*(X_{n+1})\}.$$

Theorem 3 shows that if the true domain probabilities $\pi(X)$ were known, then coverage conditional on the covariates $X$ is equivalent to coverage conditional on the domain probability vector $\pi(X)$. This implies that our method achieves approximate conditional validity whenever the learned gating vector $\hat{\pi}(X)$ is close to the true $\pi^*(X)$. In practice, the better the MoE captures domain structure, the closer we approach conditional coverage.

## 4 NUMERICAL EXPERIMENTS

In this section, we conduct synthetic data and real-data experiments to demonstrate the performance of the proposed method. Due to space constraints, details of the experiments and comprehensive results are provided in the Appendix A.4.

### 4.1 SYNTHETIC DATA EXAMPLE

The synthetic data $(X, Y)$ are drawn from a three-domain distribution:

$$Y = \begin{cases} X + \varepsilon, & p = p_1; \\ X^2 + \varepsilon, & p = p_2; \\ X^3 + \varepsilon, & p = p_3; \end{cases}$$

where $X \sim \text{Uniform}([-3, 3]), \varepsilon \sim N(0, 1)$, and $p_i, i = 1, 2, 3$, are the proportions of each domain that sum up to 1. We consider two settings: (1) Unbalanced proportions: $p_1 = 0.2, p_2 = 0.3, p_3 = 0.5$; (2) Balanced proportions: $p_1 = p_2 = p_3 = 1/3$. The sample size for training, calibration, and testing are set equally to be $n = 500$. The experiments are repeated over 50 times. The miscoverage level is set to be $\alpha = 0.1$. We denote the MoE weighted conformal prediction as MoECP. The divergence measure in MoECP is the KL divergence. The number of experts $K$ and the temperature $\tau$ for MoECP are set to be $K = 3$ and $\tau = 150$, respectively. Several state-of-the-art methods are compared, including CQR (Romano et al., 2019), CHR (Sesia & Romano, 2021), SCP+CC (Gibbs et al., 2025), RLCP (Hore & Barber, 2025), and PCP (Zhang & Candès, 2024). Details of the experimental setup are provided in the Appendix A.4.1.

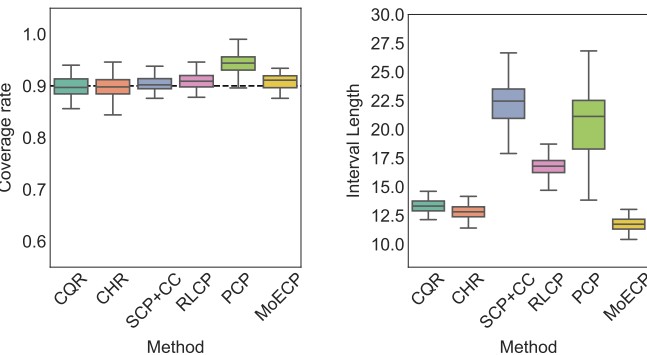

Figure 2: Marginal coverage and interval length for the setting 1 (unbalanced proportions) in synthetic data over 50 experiments.

Here, we report results for the unbalanced setting; results for the balanced setting appear in the Appendix A.4.2. Figure 2 shows marginal coverage and interval length for competing methods. All methods satisfy the coverage guarantee, but PCP is conservative. Our method, MoECP, yields the shortest prediction intervals while maintaining coverage. Notably, although MoECP is based on mean regression, it outperforms two quantile-regression–based approaches (CQR and CHR), even though quantile regression is typically more adaptive to heterogeneity. In addition, MoECP is more efficient than the mean-regression–based methods SCP+CC, RLCP, and PCP.

To highlight the differences among methods, Figure 3 presents the conditional coverage for competing approaches in a representative experiment. The coverage at each data point is computed by averaging over its 100 nearest neighbors. Among all methods, MoECP achieves conditional coverage closest to the nominal level. Figure 4 further displays the corresponding prediction intervals.

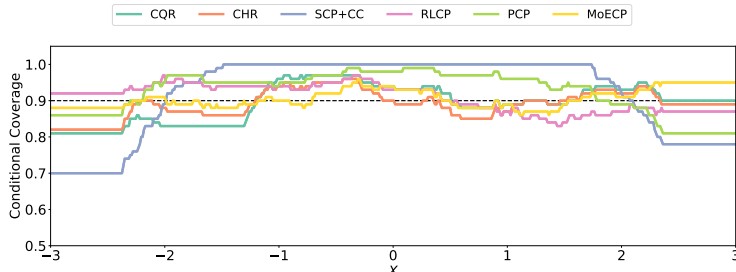

Figure 3: Local average coverage rates of conformal intervals for the setting 1 (unbalanced proportions) in synthetic data. The local coverage of a data point is average by its 100 nearest data points.

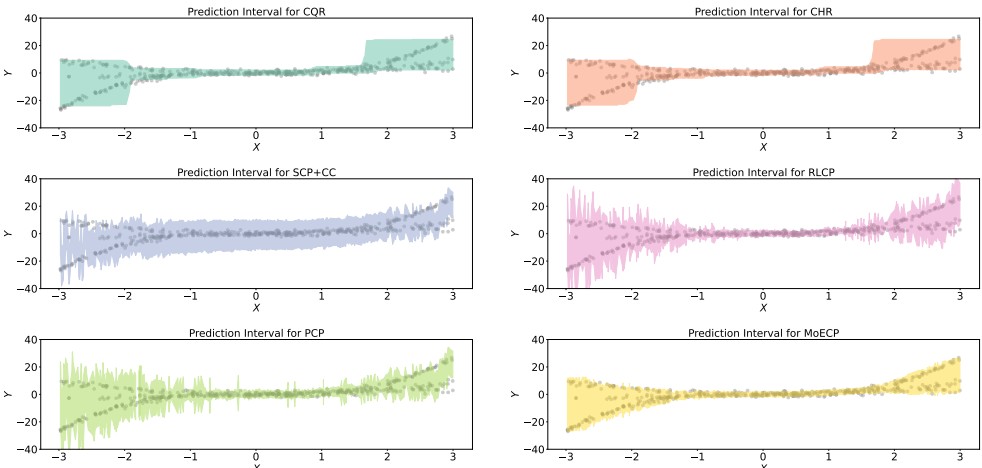

Figure 4: Prediction intervals for CQR, CHR, SCP+CC, RLCP, PCP, and MoECP for the setting 1 (unbalanced proportions) in synthetic data.

Compared with MoECP, SCP+CC, RLCP, and PCP produce noticeably wider intervals. Although CQR and CHR attain interval lengths comparable to MoECP (see Figure 2), their prediction intervals fail to adapt when substantial domain shifts occur, i.e., $X \in [-3, -2] \cup [2, 3]$. In contrast, MoECP adapts effectively across all regions, achieving both adaptivity and efficiency.

## 4.2 REAL-DATA ANALYSIS

We evaluate methods on two public datasets. The Bike-sharing dataset contains 17,389 samples with 13 features (temporal indicators such as year/month/holiday and environmental variables including weather, humidity, and wind speed); the response is rental count. The Temperature dataset comprises 7,750 summer observations (2013–2017) used to evaluate bias correction for LDAPS temperature forecasts over Seoul.

For both datasets, the training, calibration, and test sets are each fixed at $n = 1500$, and the nominal miscoverage level is set to $\alpha = 0.1$. We evaluate methods using three metrics: marginal coverage, interval length, and worst-slice coverage (Romano et al., 2020; Cauchois et al., 2021). Figure 5 reports the results. RLCP and PCP are slightly conservative, while the remaining methods attain the nominal marginal coverage. Worst-slice coverage is approximately $0.90$ for all methods, indicating satisfactory conditional coverage. Importantly, MoECP produces the shortest prediction intervals, demonstrating superior efficiency relative to the competing approaches.

It is worth noting that the MoECP framework is highly flexible, allowing us to vary the divergence measure, the number of experts, and the temperature parameter. To assess this flexibility, we conduct comprehensive ablation studies on real-world datasets (Appendix A.4.3). The results show that MoECP is robust with respect to both the divergence measure and the temperature parameter. Fur-

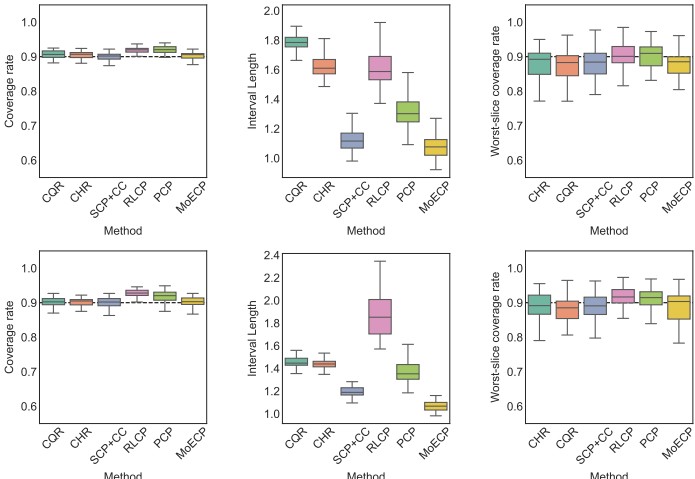

Figure 5: Marginal coverage, interval length and worst-slice coverage over 50 experiments. The first row is the results for the Bike-sharing dataset; the second row is the results for the Temperature dataset. From left to right, each column corresponds to marginal coverage, interval length, and worst-slice coverage, respectively.

thermore, when the sample size is sufficiently large, MoECP also exhibits robustness to the choice of the number of experts.

## 5 RELATED WORK

Conformal prediction (CP) provides a distribution-free framework for uncertainty quantification, producing prediction intervals with guaranteed finite-sample coverage under minimal assumptions (Vovk et al., 2005). The split conformal method calibrates a predictor by computing nonconformity scores on a held-out set and forming intervals from quantiles of these scores (Lei et al., 2018; Romano et al., 2019). Its simplicity and rigorous guarantees have led to wide adoption across fields such as medicine, natural language processing, and drug discovery (Cortés-Ciriano & Bender, 2020).

A key limitation is that standard CP ensures only marginal coverage: intervals contain the true label with the desired probability on average, but coverage can vary across subgroups or regions of the feature space (Cauchois et al., 2020; Tibshirani et al., 2019; Gibbs & Candes, 2021). This gap is problematic in heterogeneous data, where certain populations may be undercovered.

To address this, several extensions have been developed. Attribute-based methods partition the calibration data by known labels to achieve label-conditional validity (Boström et al., 2021). Localized conformal prediction assigns more weight to calibration points close to the test instance, improving adaptivity to local structure (Guan, 2023; Zhang & Candès, 2024; Hore & Barber, 2025). Another line of work considers distribution shifts, where covariate-weighted conformal prediction adjusts calibration to maintain validity under shifted test distributions (Sugiyama et al., 2007; Tibshirani et al., 2019; Candès et al., 2021; Chernozhukov et al., 2021b). Recently, several works have leveraged generative models to construct adaptive prediction intervals (Colombo, 2024; Fang et al., 2025).

Across these developments, a common scene emerges: heterogeneous data require conformal prediction methods that adapt conformity score to each example. Our work builds on this insight, introducing a MoE perspective that leverages domain similarity.

## 6 CONCLUSION

In this paper, we proposed Mixture-of-Experts Distributed Conformal Prediction (MoE-CP), a framework that leverages MoE gating vectors to construct adaptive and interpretable prediction

intervals for heterogeneous and multi-domain data. MoE-CP demonstrates that combining expert specialization with conformal calibration yields a principled and practical path toward reliable uncertainty quantification in complex data environments. By producing intervals that adapt to heterogeneous subpopulations, MoE-CP may help improve fairness and transparency in real-world application. We assume that all observations are i.i.d. samples from an overall distribution $P$, which can be modeled as a mixture of $K$ latent domains; this guarantees the exchangeability condition needed for the finite-sample marginal validity. Extending MoE-CP to settings with known domain labels or to more general non-exchangeable regimes is an interesting direction for future work. Looking forward, extending to various conformity scores and scaling to large MoE architectures opens new paths for trustworthy uncertainty quantification in real-world applications.

ACKNOWLEDGMENTS

We would like to thank the anonymous reviewers and area chairs for their great feedback on the paper. Jingsen Kong was supported by Jinan University. Wenlu Tang was supported by NSERC – Discovery Grants and Alberta Innovates. Guangren Yang's research was supported by the National Social Science Fund of China grant (24BTJ070). Bei Jiang and Linglong Kong were partially supported by grants from the Canada CIFAR AI Chairs program, the Alberta Machine Intelligence Institute (AMII), and Natural Sciences and Engineering Council of Canada (NSERC), and Linglong Kong was also partially supported by grants from the Canada Research Chair program from NSERC.

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

# A APPENDIX

Appendix A.1 outlines the full pipeline of the MoE-weighted conformal prediction procedure. Appendix A.2 contains the theoretical proofs. Appendix A.3 provides detailed guidance on hyper-parameter selection. Appendix A.4 describes the experimental setup and results, including implementation details (Appendix A.4.1), additional results for the unbalanced setting in synthetic data (Appendix A.4.2), and ablation studies on the real-data example (Appendix A.4.3). Appendix A.5 discusses the computational complexity analysis of MoECP. Appendix A.6 discusses the extension of MoECP to the online setting.

## A.1 ALGORITHM

---

**Algorithm 2** MoE-Weighted Conformal Prediction

---

**Require:** Training set $\mathcal{D}_{\text{train}} = \{(X_i, Y_i), i \in \mathcal{I}_{tra}\}$ and calibration set $\mathcal{D}_{\text{cal}} = \{(X_i, Y_i), i \in \mathcal{I}_{cal}\}$, test point $X_{n+1}$, nonconformity score function $S(\cdot, \cdot)$, number of experts $K$, divergence $D(\cdot, \cdot)$, temperature $\tau$, desired level $1 - \alpha$.
**Ensure:** MoE weighted conformal prediction interval $\hat{\mathcal{C}}_n^{\text{MoE}}(X_{n+1})$

 1: **Step 1: Train MoE model**
 2:     Initialize expert functions $\{\mu_k(X)\}_{k=1}^K$ and gating logits $\{\ell_k(X)\}_{k=1}^K$
 3:     Define gating probabilities $\pi_k(X) = \frac{\exp(\ell_k(X))}{\sum_{j=1}^K \exp(\ell_j(X))}$
 4:     Define MoE prediction $\hat{\mu}(X) = \sum_{k=1}^K \pi_k(X)\mu_k(X)$
 5:     Optimize parameters by minimizing loss function on training set $\mathcal{D}_{\text{train}}$.

 6: **Step 2: Compute nonconformity scores**
 7: **for** $i \in \mathcal{I}_{cal}$ **do**
 8:     Compute nonconformity score $S_i := S(X_i, Y_i)$ and gating vector $\pi(X_i)$
 9: **end for**

10: **Step 3: Compute weights for test point** $X_{n+1}$
11: Compute gating vector $\pi(X_{n+1})$
12: Sample $\tilde{L} \sim \text{Multinomial}(\tau, \pi(X_{n+1}))$
13: Compute $\tilde{\pi}(X_{n+1}) = \tilde{L}/\tau$
14: **for** $i \in \mathcal{I}_{cal} \cup \{n+1\}$ **do**
15:     Compute randomized weight:

$$w_i = \exp\left[-\tau D\left(\tilde{\pi}\left(X_{n+1}\right), \pi\left(X_i\right)\right)\right],$$

16: **end for**

17: **Step 4: Normalize weights**
18: **for** $i \in \mathcal{I}_{cal} \cup \{n+1\}$ **do**
19:     $\tilde{w}_i = \frac{w_i}{\sum_{j \in \mathcal{I}_{cal} \cup \{n+1\}} w_j}$
20: **end for**

21: **Step 5: Construct weighted conformal interval**
22: Form weighted empirical distribution: $\sum_{i \in \mathcal{I}_{cal}} \tilde{w}_i \delta_{S_i} + \tilde{w}_{n+1} \delta_{+\infty}$
23: Let $Q_{1-\alpha}$ be the $(1 - \alpha)$ quantile of this weighted distribution
24: **return** $\hat{\mathcal{C}}_n^{\text{MoE}}(X_{n+1}) = \{y \in \mathbb{R} : S(X_{n+1}, y) \le Q_{1-\alpha}\}$

---

## A.2 THEORETICAL PROOF

**Proof of Theorem 1.** Denote $Z_i = (X_i, Y_i), i \in [n+1]$, and $z_i$ be the realization of $Z_i$, for $i \in [n+1]$. By exchangeability,

$$\mathbb{P}\{Z_{n+1} = z_i\} = \frac{1}{n+1}, \quad i \in [n+1].$$

Conditional on $\tilde{\pi}(X_{n+1}) = \tilde{L}/\tau$, and using the Bayes theorem, we have

$$\mathbb{P}\{Z_{n+1} = z_i \mid \tilde{\pi}(X_{n+1})\} = \frac{\mathbb{P}\{\tilde{\pi}(X_{n+1}) \mid Z_{n+1} = z_i\}\mathbb{P}\{Z_{n+1} = z_i\}}{\sum_{j=1}^{n+1} \mathbb{P}\{\tilde{\pi}(X_{n+1}) \mid Z_{n+1} = z_j\}\mathbb{P}\{Z_{n+1} = z_j\}}$$

$$= \frac{\prod_{k=1}^{K}[\pi_k(x_i)]^{\tilde{L}_k}}{\sum_{j=1}^{n+1} \prod_{k'=1}^{K}[\pi_{k'}(x_j)]^{\tilde{L}_{k'}}}$$

$$= \tilde{w}_i, \quad i \in [n+1],$$

Here,

$$\prod_{k=1}^{K}[\pi_k(x_i)]^{\tilde{L}_k} = \exp\left(\sum_{k=1}^{K} \tilde{L}_k \log \pi_k(x_i)\right)$$

$$= \exp\left(\tau \sum_{k=1}^{K} \tilde{\pi}_k(x_{n+1}) \log \pi_k(x_i)\right)$$

$$= \exp\left(-\tau H(\tilde{\pi}(x_{n+1}), \pi(x_i))\right),$$

where $H(\cdot, \cdot)$ is Cross Entropy. Likewise, dividing both the numerator and denominator by the same number does not change the result. We know that

$$\prod_{k=1}^{K}\left[\frac{\pi_k(x_i)}{\tilde{\pi}_k(x_{n+1})}\right]^{\tilde{L}_k} = \exp\left(-\tau KL(\tilde{\pi}(x_{n+1})\|\pi(x_i))\right),$$

where $KL(\cdot\|\cdot)$ is the KL divergence. Note that

$$Y_{n+1} \notin \hat{\mathcal{C}}_n^{\text{MoE}}(X_{n+1}) \iff S_{n+1} \geq Q_{1-\alpha}\left(\sum_{i=1}^{n} \tilde{w}_i \delta_{S_i} + \tilde{w}_{n+1}\delta_\infty\right)$$

$$\iff S_{n+1} \geq Q_{1-\alpha}\left(\sum_{i=1}^{n+1} \tilde{w}_i \delta_{S_i}\right) \quad \text{(Lemma A.1 in Guan (2023))}$$

$$\iff \sum_{i=1}^{n+1} \tilde{w}_i I(S_i \geq S_{n+1}) \leq \alpha$$

Therefore, for the MoE weighted conformal prediction with Cross Entropy or KL divergence-based weights, the probability conditional on $\tilde{\pi}(X_{n+!})$ is

$$\mathbb{P}\{Y_{n+1} \notin \hat{\mathcal{C}}_n^{\text{MoE}}(X_{n+1}) \mid \tilde{\pi}(X_{n+1})\}$$

$$= \sum_{j=1}^{n+1} \mathbb{P}\{Y_{n+1} \notin \hat{\mathcal{C}}_n^{\text{MoE}}(X_{n+1}) \mid \tilde{\pi}(X_{n+1}), Z_{n+1} = z_j\} \times \mathbb{P}\{Z_{n+1} = z_j \mid \tilde{\pi}(X_{n+1})\}$$

$$= \sum_{j=1}^{n+1} \tilde{w}_j \mathbb{E}\left[I\left(\sum_{i=1}^{n+1} \tilde{w}_i I(S_i \geq S_j) \leq \alpha\right)\right]$$

$$= \mathbb{E}\left[\sum_{j=1}^{n+1} \tilde{w}_j I\left(\sum_{i=1}^{n+1} \tilde{w}_i I(S_i \geq S_j) \leq \alpha\right)\right]$$

$$\leq \mathbb{E}[\alpha]$$

$$= \alpha,$$

where $I(\cdot)$ is the indicator function, and the relationship "$\leq$" holds due to Lamma A1 in Harrison (2012). Therefore,

$$\mathbb{P}\{Y_{n+1} \in \hat{\mathcal{C}}_n^{\text{MoE}}(X_{n+1}) \mid \tilde{\pi}(X_{n+1}\} \geq 1 - \alpha.$$

**Proof of Theorem 2.** By Assumption 1, for each $i$, there exists $r_i = o(1)$ with

$$\tilde{w}_{i,D} = \tilde{w}_{i,KL} + r_i, \text{ and } \sum_{i=1}^{n+1} r_i = 0.$$

Define

$$\hat{F}_{n+1}^{(D)}(s) = \sum_{i=1}^{n+1} \tilde{w}_{i,D} \cdot I(S_i \leq s), \text{ and } \hat{F}_{n+1}^{(KL)}(s) = \sum_{i=1}^{n+1} \tilde{w}_{i,KL} \cdot I(S_i \leq s).$$

For any fixed $s$,

$$|\hat{F}_{n+1}^{(D)}(s) - \hat{F}_{n+1}^{(KL)}(s)| \leq \sum_{i=1}^{n+1} |r_i| = o(1).$$

Then, $\|\hat{F}^{(D)} - \hat{F}^{(KL)}\|_\infty = o(1)$.

Let

$$q_{1-\alpha}^{(D)} = \inf\{s : \hat{F}_{n+1}^{(D)}(s) \geq 1 - \alpha\}, \text{ and } q_{1-\alpha}^{(KL)} = \inf\{s : \hat{F}_{n+1}^{(KL)}(s) \geq 1 - \alpha\}.$$

As $S$ is continuous, we have

$$q_{1-\alpha}^{(D)} = q_{1-\alpha}^{(KL)} + o(1).$$

The MoE weighted conformal interval with similarity measure $D$ is

$$\hat{\mathcal{C}}_n^{(D)}(X_{n+1}) = \{y : S(X_{n+1}, y) \leq q_{1-\alpha}^{(D)}\}.$$

Likewise, the interval with KL divergence is

$$\hat{\mathcal{C}}_n^{(KL)}(X_{n+1}) = \{y : S(X_{n+1}, y) \leq q_{1-\alpha}^{(KL)}\}.$$

We know that

$$\mathbb{P}\{Y_{n+1} \in \hat{\mathcal{C}}_n^{KL}(X_{n+1}) \mid \tilde{\pi}(X_{n+1})\} = \mathbb{P}\{S_{n+1} \leq q_{1-\alpha}^{(KL)} \mid \tilde{\pi}(X_{n+1})\} \geq 1 - \alpha.$$

Thus,

$$\begin{aligned}
\mathbb{P}\{Y_{n+1} \in \hat{\mathcal{C}}_n^{(D)}(X_{n+1}) \mid \tilde{\pi}(X_{n+1})\} &= \mathbb{P}\{S_{n+1} \leq q_{1-\alpha}^{(D)} \mid \tilde{\pi}(X_{n+1})\} \\
&= \mathbb{P}\{S_{n+1} \leq q_{1-\alpha}^{(KL)} + o(1) \mid \tilde{\pi}(X_{n+1})\} \\
&\geq 1 - \alpha + o(1),
\end{aligned}$$

which completes the proof.

**Proof of Theorem 3.** The proof is somewhat similar to Proposition 1 in Zhang & Candès (2024), while they consider a different problem setting. Note that

$$\tilde{w}_i^* = \frac{\exp[-\tau D(\pi^*(X_{n+1}), \pi^*(X_i))]}{\sum_{i=1}^{n+1} \exp[-\tau D(\pi^*(X_{n+1}), \pi^*(X_i))]}, \quad i \in [n+1].$$

Therefore, for any similarity measure $D(\cdot, \cdot)$ and $\tau \in \mathbb{N}^+$, we have $\tilde{w}_1^*, \cdots, \tilde{w}_{n+1}^* \perp X_{n+1} \mid \pi^*(X_{n+1})$. By the construction of $\hat{\mathcal{C}}_n^{\text{MoE},*}(X_{n+1})$,

$$Y_{n+1} \notin \hat{\mathcal{C}}_n^{\text{MoE},*}(X_{n+1}) \iff \hat{P}_n^* := \sum_{i=1}^{n+1} \tilde{w}_i^* I(S_i \geq S_{n+1}) \leq \alpha,$$

which means that the event $\{Y_{n+1} \notin \hat{\mathcal{C}}_n^{\text{MoE},*}(X_{n+1})\}$ depends on $X_{n+1}$ only through $S_{n+1}$ and $\pi^*(X_{n+1})$ in the weights $\tilde{w}_i^*, i \in [n+1]$. If there is a finite mixture representation such that $S_i | X_i \sim \sum_{k=1}^K \pi_k^*(X_i) f_k^*, i \in [n+1]$, we have

$$S_{n+1} \perp X_{n+1} \mid \pi^*(X_{n+1}).$$

Therefore, the probability of the event given $\pi^*(X_{n+1})$ and $X_{n+1}$ is independent of $X_{n+1}$. i.e.,

$$\mathbb{P}\{\hat{P}_n^* \le \alpha \mid X_{n+1}, \pi^*(X_{n+1})\} = \mathbb{P}\{\hat{P}_n^* \le \alpha \mid \pi^*(X_{n+1})\}.$$

Conversely, $\pi(X_{n+1})$ can be taken as a function of $X_{n+1}$. Once $X_{n+1}$ is given, $\pi(X_{n+1})$ can be taken as fixed. Then,

$$\mathbb{P}\{\hat{P}_n^* \le \alpha \mid X_{n+1}, \pi^*(X_{n+1})\} = \mathbb{P}\{\hat{P}_n^* \le \alpha \mid X_{n+1}\}.$$

Consequently,

$$\mathbb{P}\{\hat{P}_n^* \le \alpha \mid \pi^*(X_{n+1})\} = \mathbb{P}\{\hat{P}_n^* \le \alpha \mid X_{n+1}\}.$$

Taking the complement on both sides completes the proof.

**Proposition related to Assumption 1.** Here, we state a proposition that Assumption 1 is satisfied under a specified condition. This condition satisfies many divergence measures if there are enough calibration points. The rationale behind the claim is that, for a given $\tilde{\pi} := \tilde{\pi}(x_{n+1})$, if the calibration set is sufficiently large and contains a point whose gating vector $\pi_j := \pi(x_j)$ equals (or arbitrarily closely approximates) $\tilde{\pi}$, then the KL divergence satisfies $KL(\tilde{\pi}, \pi_j) \approx 0$. Since $KL(\tilde{\pi}, q) = 0$ if and only if $q = \tilde{\pi}$, such a calibration point attains the minimizer of the KL divergence. For any divergence $D$ that likewise vanishes only at equality (for example, Jeffreys divergence, the Hellinger distance, or the Euclidean distance, etc), we have $D(\tilde{\pi}, \pi_j) = 0$. Hence, the minimizer condition is satisfied.

**Proposition 1** *If a divergence $D(\cdot, \cdot)$ shares the same minimizer as with the KL divergence, then Assumption 1 is verified, i.e., $\tilde{w}_{i,D} = \tilde{w}_{i,KL} + o(1)$, as $\tau \to \infty$.*

**Proof of Proposition 1.** If the divergence $D$ shares the same unique minimizer with KL divergence, we have

$$i^* = \arg\min_{i \in [n+1]} \tilde{w}_{i,D} = \arg\min_{i \in [n+1]} \tilde{w}_{i,KL}.$$

Let

$$f_i = D(\tilde{\pi}, \pi_i), \quad g_i = KL(\tilde{\pi}, \pi_i), \quad i \in [n+1],$$

where $\tilde{\pi} := \tilde{\pi}(x_{n+1}), \pi_i := \pi(x_i)$. Denote

$$f_{(1)} = \min_i f_i, \quad f_{(2)} = \min_{i: f_i > f_{(1)}} f_i,$$

similar definition for $g_{(1)}, g_{(2)}$. Suppose that there exists the same minimizer, i.e.,

$$f_{i^*} = f_{(1)}, \quad g_{i^*} = g_{(1)}.$$

We define

$$\delta := \min\{f_{(2)} - f_{i^*}, g_{(2)} - g_{i^*}\} > 0.$$

Then, the normalized weights for the divergence $D$ and KL can be expressed as

$$\tilde{w}_i^{(f)}(\tau) := \frac{e^{-\tau f_i}}{\sum_{j=1}^{n+1} e^{-\tau f_j}}, \quad \tilde{w}_i^{(g)}(\tau) := \frac{e^{-\tau g_i}}{\sum_{j=1}^{n+1} e^{-\tau g_j}}.$$

We can rewrite the denominator as

$$S_f := \sum_{j=1}^{n+1} e^{-\tau f_j} = e^{-\tau f_{i^*}}(1 + A_f), \quad S_g := \sum_{j=1}^{n+1} e^{-\tau g_j} = e^{-\tau g_{i^*}}(1 + A_g),$$

where $A_f := \sum_{j \ne i^*} e^{-\tau(f_j - f_{i^*})}, A_g := \sum_{j \ne i^*} e^{-\tau(g_j - g_{i^*})}$. Clearly,

$$0 \le A_f, A_g \le n e^{-\tau \delta}.$$

In case $i = i^*$, we know that

$$\tilde{w}_{i^*}^{(f)} = \frac{e^{-\tau f_{i^*}}}{S_f} = \frac{1}{1 + A_f}, \quad \tilde{w}_{i^*}^{(g)} = \frac{e^{-\tau g_{i^*}}}{S_g} = \frac{1}{1 + A_g}.$$

Therefore,

$$|\tilde{w}_{i^*}^{(f)} - \tilde{w}_{i^*}^{(g)}| = \frac{|A_g - A_f|}{(1 + A_f)(1 + A_g)} \le |A_g - A_f| \le \max\{A_g, A_f\} \le ne^{-\tau\delta}, \quad (6)$$

In case $i \ne i^*$, we know that

$$w_i^{(f)} = \frac{e^{-\tau f_i}}{S_f} = \frac{e^{-\tau(f_i - f_{i^*})}}{1 + A_f}, \quad w_i^{(g)} = \frac{e^{-\tau g_i}}{S_g} = \frac{e^{-\tau(g_i - g_{i^*})}}{1 + A_g}.$$

Let $\Delta f_i := f_i - f_{i^*}, \Delta g_i := g_i - g_{i^*}$. We know that $\Delta f_i \ge \delta, \Delta g_i \ge \delta$, for $i \ne i^*$. Then, the absolute difference between two weights can be decomposed as

$$
\begin{aligned}
|\tilde{w}_i^{(f)} - \tilde{w}_i^{(g)}| &= \left| \frac{e^{-\tau\Delta f_i}}{1 + A_f} - \frac{e^{-\tau\Delta g_i}}{1 + A_g} \right| \\
&\le \left| \frac{e^{-\tau\Delta f_i} - e^{-\tau\Delta g_i}}{1 + A_f} \right| + e^{-\tau\Delta g_i} \left| \frac{1}{1 + A_f} - \frac{1}{1 + A_g} \right| \\
&\le |e^{-\tau\Delta f_i} - e^{-\tau\Delta g_i}| + e^{-\tau\Delta g_i}|A_g - A_f| \\
&\le \max\{e^{-\tau\Delta f_i}, e^{-\tau\Delta g_i}\} + e^{-\tau\Delta g_i} \max\{A_g, A_f\} \\
&\le e^{-\tau\delta} + ne^{-2\tau\delta}.
\end{aligned}
\quad (7)
$$

Based on Equation (6) and (7), we can conclude that

$$\tilde{w}_{i,D} = \tilde{w}_{i,KL} + o(1),$$

as $\tau \to \infty$, if the divergence $D$ shares the same the minimizer with the KL divergence.

### A.3 CHOICE OF HYPERPARAMETERS

The MoECP framework allows the user to specify the divergence measure, the number of experts $K$, and the temperature $\tau$. The ablation studies (Appendix A.4.3) provide a detailed comparison among these hyperparameters. Here, we provide practical guidance with respect to these choices.

**Divergence measure** $D(\cdot, \cdot)$. We demonstrate both theoretically (Theorem 2) and empirically (Appendix A.4.3) that MoECP is largely insensitive to the particular divergence measure employed. In addition, we prove that the KL divergence enjoys exact marginal validity within our framework. For these reasons, combining theoretical guarantees and empirical robustness, we recommend the KL divergence as the default choice for MoECP in practice.

**Numbers of experts** $K$. Our ablation studies show that whether $K$ materially affects MoECP's interval length depends primarily on the training and calibration sample size. Intuitively, increasing $K$ partitions the calibration data across more experts and thus reduces the effective sample size per expert; when the total sample size is modest (for example, $n \le 1500$), this reduction can noticeably widen prediction intervals, so a smaller $K$ (e.g., $K = 2, 3$) is preferable. For larger overall sample sizes, each expert still receives sufficient data and the difference in interval length across different $K$ becomes negligible. In practice, choose $K$ by balancing statistical efficiency and modeling flexibility: use cross-validation or information criteria (BIC/AIC) when possible, taking into account domain knowledge and computational constraints. As an alternative strategy, one may fit an over-parameterized model and apply sparsity regularization on the gating network to automatically prune unused experts, thereby producing an effective $K$ without exhaustive manual tuning.

**Temperature parameter** $\tau$. Our ablation studies (with $\tau$ scanned in the range 50–300) indicate that MoECP is robust to the choice of temperature: marginal coverage, conditional coverage, and interval length remain stable across a wide range of $\tau$. Intuitively, $\tau$ controls the sharpness of the MoE weights—very small $\tau$ produces nearly uniform weights and reduces adaptivity, while very large $\tau$ concentrates weight on a few points (reducing the effective calibration sample size and widening intervals). A moderate $\tau$ therefore provides a good trade-off between adaptivity and statistical efficiency; we recommend $\tau \in [100, 300]$ as a practical default. When needed, $\tau$ can be tuned by simple grid search or cross-validation.

## A.4 EXPERIMENT SETUP AND RESULTS

### A.4.1 DETAILS OF THE EXPERIMENTAL SETUP

We use the *Mixture of Experts (MoE)* predictor as the base estimator for the proposed MoECP and implement an MoE model with $K$ experts and a single gating network. Each expert is a fully connected feedforward network with three hidden layers and ReLU activations; the gating network uses the same hidden-layer structure and outputs a softmax distribution over the $K$ experts to weight their predictions. A dropout layer with rate $0.1$ is applied in both the experts and the gating network to reduce overfitting. Models were trained with mini-batch gradient descent (batch size $64$), a learning rate of $1 \times 10^{-4}$, and an early stopping ceiling on epochs. For the synthetic-data experiments we set $K = 3$, temperature $\tau = 150$, use hidden layers of size $(32, 32, 32)$ for both experts and gate, and train up to $600$ epochs. For the real-data experiments we set $K = 2$, temperature $\tau = 100$, use hidden layers of size $(64, 64, 64)$, and train up to $2000$ epochs.

For Conformalized quantile regression (CQR) (Romano et al., 2019) and Conformalized histogram regression (CHR) (Sesia & Romano, 2021), we used the original authors' implementation using the base estimator as a quantile random forest. [1]. For Split conformal prediction with conditional calibration (SCP+CC) (Gibbs et al., 2025), Randomly-localized conformal prediction (RLCP) (Hore & Barber, 2025), and Posterior conformal prediction (PCP) (Zhang & Candès, 2024), we use the implementation in Zhang & Candès (2024) using the base estimator as a random forest regression[2], which use `sklearn.ensemble.RandomForestRegressor` in `scikit-learn` with default hyperparameter. Specifically, the bandwidth of RLCP is chosen such that the effective sample size is 100 to balance the trade-off between the adaptivity and efficiency (Hore & Barber, 2025)).

To evaluate conditional coverage in the real-data experiments, we adopt the *worst-slice coverage* metric, following Romano et al. (2020); Cauchois et al. (2021). This metric approximates the worst coverage across all small subpopulations of the form $\{x : v^{\top} x \in [a, b]\}$, where $a$, $b$, and $v$ may vary arbitrarily.

The Bike-sharing dataset (UCI Machine Learning Repository)[3] comprises 17,389 observations from an urban bike-sharing system. Each record includes 13 explanatory features that capture temporal context (e.g., year, month, weekday, holiday indicators, and working-day flags) and environmental conditions (e.g., ambient temperature, perceived temperature, humidity, wind speed, and a coarse weather category). The response is the total rental count. Categorical and ordinal variables were encoded appropriately (binary flags or integer codes), and all numeric predictors and the response were standardized to zero mean and unit variance prior to model training, calibration, and evaluation.

The Temperature dataset (UCI Machine Learning Repository)[4] comprises 7,750 summer observations (2013–2017) assembled to evaluate bias-correction methods for LDAPS numerical temperature forecasts over the Seoul region. Each record pairs LDAPS forecast outputs with the corresponding ground-truth observation and includes temporal indicators (e.g., date/time, day-of-year) and local meteorological covariates used for bias modeling. In our experiments we use the next day's minimum temperature as the response; we drop the fields `station`, `Date`, and `Next_Tmax`, and retain the remaining 21 variables as predictors. All variables - both features and response - are standardized to zero mean and unit variance prior to training, calibration, and evaluation.

### A.4.2 ADDITIONAL RESULTS IN SYNTHETIC DATA EXAMPLES

In section 4.1, the synthetic data $(X, Y)$ are drawn from a three-domain distribution:

$$Y = \begin{cases} X + \varepsilon, & p = p_1; \\ X^2 + \varepsilon, & p = p_2; \\ X^3 + \varepsilon, & p = p_3; \end{cases}$$

where $X \sim \text{Uniform}([-3, 3]), \varepsilon \sim N(0, 1)$, and $p_i, i = 1, 2, 3$, are the proportions of each domain that sum up to 1. We consider two settings: (1) Unbalanced proportions: $p_1 = 0.2, p_2 = 0.3, p_3 =$

---

[1]https://github.com/msesia/chr

[2]https://github.com/yaozhang24/pcp

[3]https://archive.ics.uci.edu/dataset/275/bike+sharing+dataset

[4]https://archive.ics.uci.edu/dataset/514/bias+correction+of+numerical+prediction+model+temperature+forecast

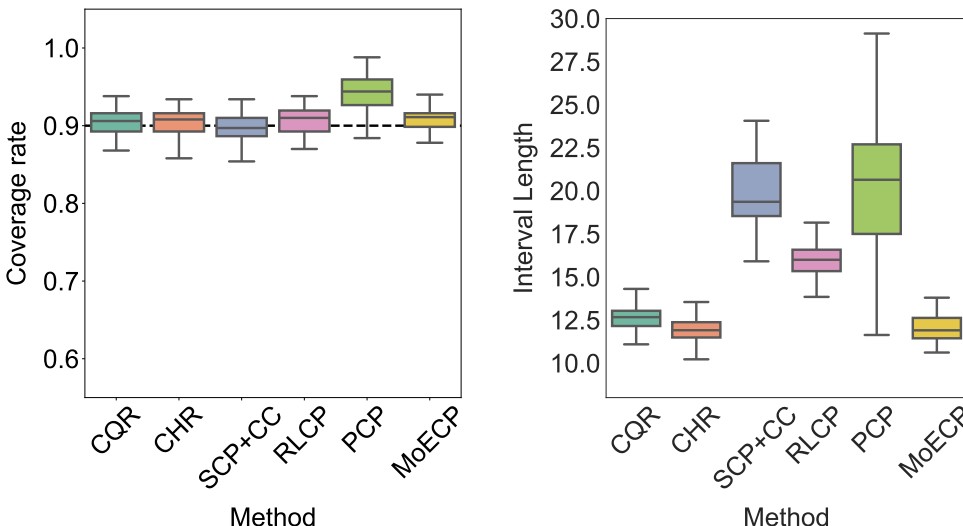

Figure 6: Marginal coverage and interval length for the setting 2 (balanced proportions) in synthetic data over 50 experiments.

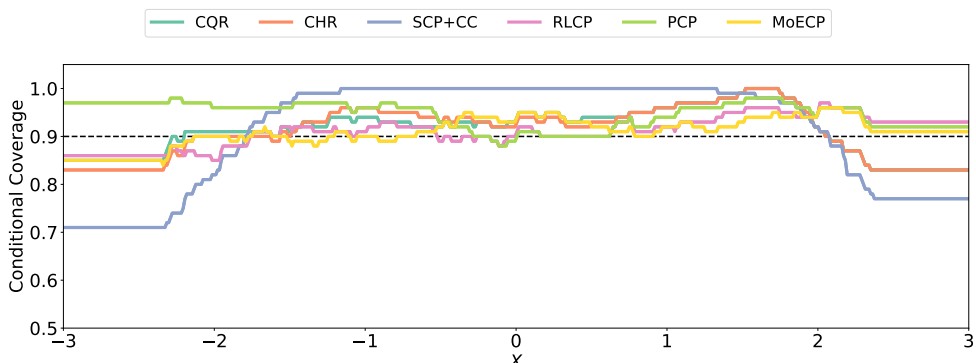

Figure 7: Local average coverage rates of conformal intervals in the setting 2 (balanced proportion) in synthetic data. The local coverage of a data point is average by its 100 nearest data points.

0.5; (2) Balanced proportions: $p_1 = p_2 = p_3 = 1/3$. The sample size for training, calibration, and testing are set equally to be $n = 500$. The experiments are repeated over 50 times. The miscoverage level is set to be $\alpha = 0.1$.

Figures 2, 3, and 4 present the results under Setting 1 (unbalanced proportions), while Figures 6, 7, and 8 correspond to Setting 2 (balanced proportions). Across both settings, the experimental results demonstrate that the proposed MoECP framework achieves reliable marginal coverage, strong conditional coverage, and high efficiency with adaptively small interval lengths.

### A.4.3 ABLATION STUDY

The MoECP framework is flexible, allowing different choices of divergence $D(\cdot, \cdot)$, number of experts $K$, and temperature $\tau$. To examine the impact of these hyperparameters, we conduct ablation studies on the Bike-sharing and Temperature datasets. The reported results are averaged over 10 independent runs.

**Ablation studies on divergence $D(\cdot, \cdot)$ and temperature $\tau$.** We consider several divergence measures, including KL divergence, Rényi divergence, Jeffreys divergence, Hellinger distance, Euclidean distance, and Cosine distance. The temperature parameter $\tau$ is varied from 50 to 300, while

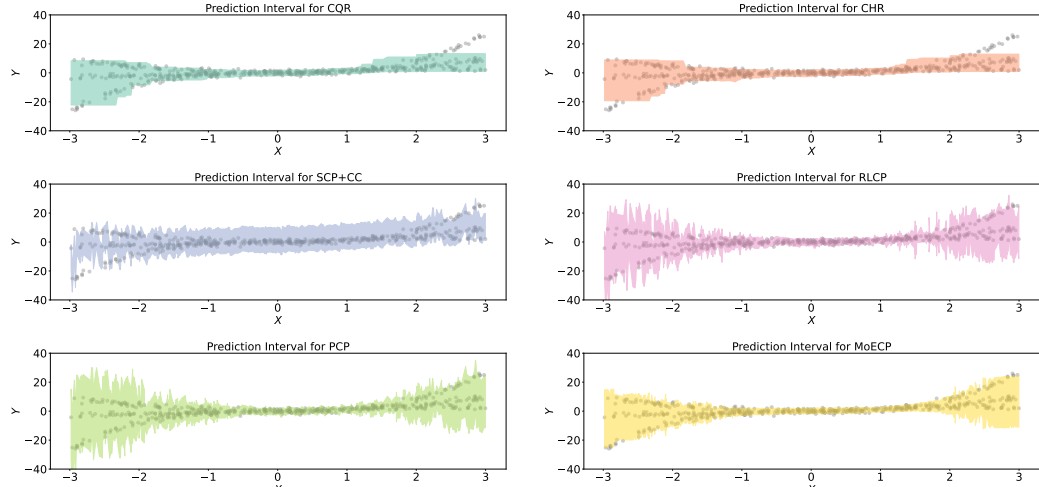

Figure 8: Prediction intervals for CQR, CHR, SCP+CC, RLCP, PCP and MoECP for the setting 2 (balanced proportions) in synthetic data.

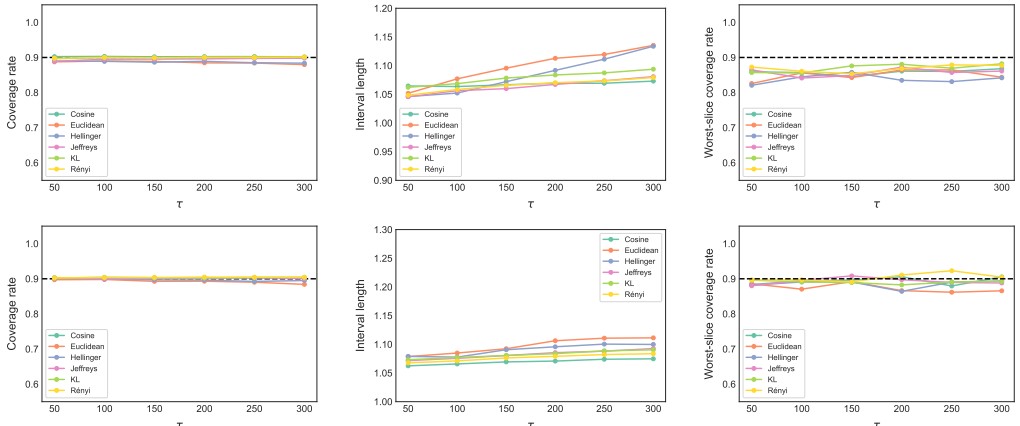

Figure 9: Ablation studies on divergence $D(\cdot, \cdot)$ and temperature parameter $\tau$. The sample size for training, calibration and testing is set equally to be $n = 1500$. The number of experts is $K = 2$. The first row is the results of the Bike-sharing dataset; the second row is the results of the Temperature dataset. From left to right, each column corresponds to marginal coverage, prediction interval length, and worst-slice coverage, respectively.

the number of experts is fixed at $K = 2$. For all experiments, the training, calibration, and testing sets are of equal size, with $n = 1500$ samples each.

Figure 9 reports the results of ablation studies over divergence measures and temperature, demonstrating the robustness of the MoECP framework with respect to these hyperparameters. In particular, the marginal coverage remains close to the nominal level across different divergences and values of $\tau$. The interval length is also largely stable, with only minor variations across divergences and temperatures. As $\tau$ increases, we observe a slight increase in interval length. This is expected, since a larger $\tau$ reduces the effective sample size, thereby widening the prediction intervals (in the extreme case $\tau \to +\infty$, the normalized weights collapse to $\tilde{w}_{n+1} = 1$ and $\tilde{w}_i = 0$ for all $i \in 1, \cdots, n$, yielding an infinite prediction interval). Moreover, the worst-slice coverage is consistently robust with respect to both divergence and temperature.

The expressions of KL divergence, Rényi divergence, Jeffreys divergence, Hellinger distance, Euclidean distance, and Cosine distance are listed below. For $\tilde{\pi}(X_{n+1}), \pi(X_i) \in [0, 1]^K$,

*KL divergence:*

$$D_{KL}(\tilde{\pi}(X_{n+1}), \pi(X_i)) = \sum_{k=1}^{K} \tilde{\pi}_k(X_{n+1}) \log \frac{\tilde{\pi}_k(X_{n+1})}{\pi_k(X_i)}.$$

*Jeffreys divergence:*

$$D_{\text{Jeffreys}}(\tilde{\pi}(X_{n+1}), \pi(X_i)) = D_{KL}(\tilde{\pi}(X_{n+1}), \pi(X_i)) + D_{KL}(\pi(X_i), \tilde{\pi}(X_{n+1})).$$

*Rényi divergence with $\alpha = 0.5$:*

$$D_{\text{Rényi}}(\tilde{\pi}(X_{n+1}), \pi(X_i)) = \frac{1}{\alpha - 1} \log \left( \sum_{k=1}^{K} \tilde{\pi}_k^{\alpha}(X_{n+1}) \pi_k^{1-\alpha}(X_{n+1}) \right).$$

*Hellinger distance:*

$$D_{\text{Hellinger}}(\tilde{\pi}(X_{n+1}), \pi(X_i)) = \frac{1}{\sqrt{2}} \left\| \sqrt{\tilde{\pi}(X_{n+1})} - \sqrt{\pi(X_i)} \right\|_2.$$

*Euclidean distance:*

$$D_{\text{Euclidean}}(\tilde{\pi}(X_{n+1}), \pi(X_i)) = \|\tilde{\pi}(X_{n+1}) - \pi(X_i)\|_2$$

*Cosine distance:*

$$D_{\text{Cosine}}(\tilde{\pi}(X_{n+1}), \pi(X_i)) = 1 - \frac{\tilde{\pi}(X_{n+1}) \cdot \pi(X_i)}{\|\tilde{\pi}(X_{n+!})\|_2 \|\pi(X_i)\|_2}.$$

Note that for the cross-entropy,

$$D_{\text{cross-entropy}}(\tilde{\pi}(X_{n+1}), \pi(X_i)) = D_{\text{KL}}(\tilde{\pi}(X_{n+1}), \pi(X_i)) - \sum_{k=1}^{K} \tilde{\pi}_k(X_{n+1}) \log \tilde{\pi}_k(X_{n+1}).$$

A direct calculation shows that the normalized weights in the MoE weighted conformal prediction satisfy

$$\tilde{w}_{KL,i} = \tilde{w}_{\text{cross-entropy},i}.$$

Therefore, it is unnecessary to include the cross-entropy in the ablation studies, although it can be used equivalently.

**Ablation studies on the experts $K$.** We further conduct ablation studies by varying the number of experts $K \in [2, 3, 4, 5]$. In addition, different sample sizes for training and calibration are considered, ranging from 500 to 2500 (e.g., $n = 1000$ means both training and calibration sets contain 1000 samples each). The test set size is fixed at $n = 1000$ throughout. The temperature is fixed at $\tau = 100$, and the divergence measure is chosen as the KL divergence.

Figure 10 presents the results of ablation studies with varying numbers of experts $K$ and different sample sizes. We observe that the marginal coverage consistently matches the nominal level across all settings, and the worst-slice coverage remains robust with respect to both $K$ and the sample size. Regarding interval length, it naturally decreases as the training and calibration sample sizes increase. When fixing the sample size and varying $K$, MoECP with smaller $K$ generally yields shorter intervals. This occurs because a smaller $K$ corresponds to a larger effective sample size, leading to narrower prediction intervals. For example, with calibration size $n = 500$ and assuming balanced domains, $K = 2$ results in 250 samples per domain, whereas $K = 5$ leaves only 100 samples per domain, thus widening the intervals. However, this effect diminishes as the overall sample size grows (e.g., $n = 3000$), since each domain still has sufficient data to construct tight prediction intervals even with larger $K$.

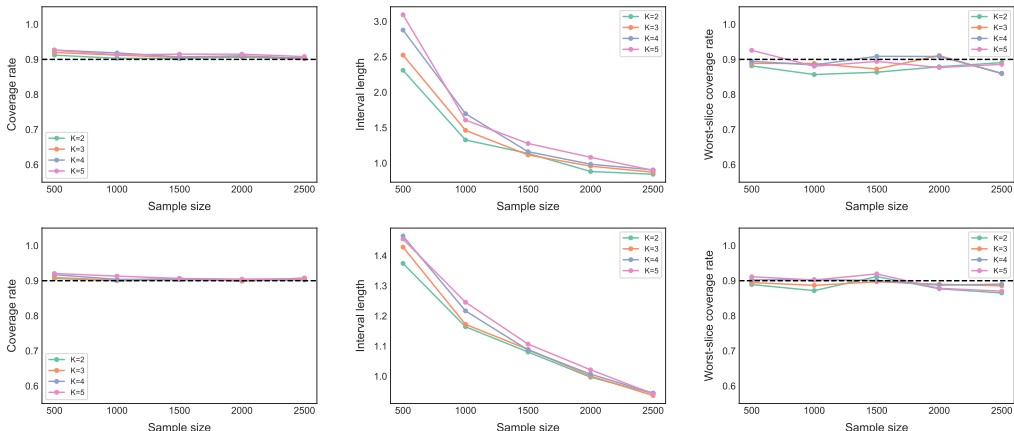

Figure 10: Ablation studies on the experts number $K$ with different sample size. The temperature parameter is set to be $\tau = 100$. The divergence measure is the KL divergence. The sample size for testing is set to be $n = 1000$, while the sample size for training and calibration varies from 500 to 2500. The first row is the results of the Bike-sharing dataset; the second row is the results of the Temperature dataset. From left to right, each column corresponds to marginal coverage, prediction interval length, and worst-slice coverage, respectively.

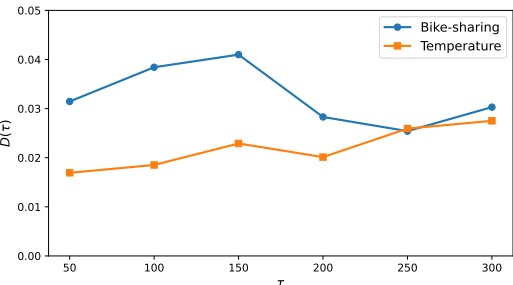

Figure 11: Ablation studies on the impact of multinomial sampling on MoECP interval variability and stability. The divergence measure is the KL divergence, and the number of experts $k = 2$. The sample size for training and calibration is set equally to be $n = 1500$. The sample size for the test set to calculate the variability measure is set to 100, and the multinomial sampling times are 500 for each test point.

**Ablation studies on the impact of multinomial sampling on MoECP interval variability and stability**  For MoECP, the randomized gating probability $\tilde{\pi}(X_{n+1})$ is generated using the multinomial sampling scheme in (3) with temperature parameter $\tau$. Our goal is to examine how this randomized construction influences the variability and stability of the resulting prediction intervals. To quantify this effect, we adopt a variability metric inspired by Hore & Barber (2025):

$$D(\tau) = \mathbb{E} \left[ \frac{\text{MAD} \left( \lambda(\hat{\mathcal{C}}_n^{\text{MoE}}(X_{n+1})) \right)}{\text{Median} \left( \lambda(\hat{\mathcal{C}}_n^{\text{MoE}}(X_{n+1})) \right)} \right],$$

where $\lambda(\cdot)$ denotes the Lebesgue measure of a prediction interval and $\text{MAD}(X) = \text{Median}(|X - \text{Median}(X)|)$. The only source of randomness comes from the multinomial sampling. Thus, $D(\tau)$ captures the degree to which the MoECP interval width fluctuates. Experiments are performed on the Bike-Sharing and Temperature datasets, with training/calibration size 1500 and test size 100. The MoE model configuration follows Section 4.2, using KL divergence and two experts ($K = 2$). For each test point, the multinomial sampling is repeated 500 times to estimate the variability.

Figure 11 presents the interval variability and stability results. The variability metric $D(\tau)$ is stable across values of the temperature parameter $\tau$ for both real datasets, indicating that the choice of $\tau$ is robust on interval variability and stability.

## A.5 COMPUTATIONAL COMPLEXITY ANALYSIS

For MoECP, it needs to train an MoE model in the training phase, and calculate the conformal score in calibration phase. Here, we aim to investigate the computational complexity of MoECP. We highlight that the dominant computational cost lies in training the MoE model, while the calibration step of MoE-CP is lightweight.

- **Training cost (dominant).** Training the MoE involves forward/backward passes through the gating network and all experts. If $m$ is the training sample size, $K$ the number of experts, and $E$ the number of epochs, the overall training cost is on the order of $O(Em(C_{\text{gate}} + KC_{\text{exp}}))$, where $C_{\text{gate}}$ and $C_{\text{exp}}$ denote one forward-backward cost of the gate and a single expert, respectively. This is a one-time offline cost and represents the main computational burden of the method. In our experiments, training is performed once per dataset and is easily handled.

- **Calibration cost (lightweight).** Once the MoE is trained, we precompute and cache the gating vectors $\pi(X_i), i \in [n]$, of the calibration set. For each test point, MoE-CP computes (1) one forward pass of the gating network, and (2) pairwise divergences between the test gating vector and the $n$ calibration gating vectors. This results in an $O(n)$ cost, which is extremely fast in practice. Actually, the computational complexity of the calibration stage is analogous to RLCP (Hore & Barber, 2025).

We evaluate the practical runtime on the real datasets, averaging each result over 10 independent runs. All experiments are conducted on an AMD Ryzen 5 4600H processor with Radeon graphics. Note that the reported runtime would be substantially lower with more advanced computing hardware. The details of the experimental setup are provided in Appendix A.4.1.

Table 1 reports the runtime for each method on the Bike-sharing and Temperature datasets. The computational cost of MoE-CP is comparable to that of CHR and PCP. Figure 12 illustrates the runtime as a function of the training/calibration sample size, showing an approximately linear increase in computational time with sample size.

Table 1: Runtime (seconds) for each method and dataset

| Methods | CQR | CHR | SCP+CC | RLCP | PCP | MoE-CP |
|---|---|---|---|---|---|---|
| Bike-sharing | 3.3 | 151.7 | 27.0 | 5.1 | 134.7 | 153.2 |
| Temperature | 3.8 | 153.4 | 39.1 | 6.4 | 157.6 | 155.5 |

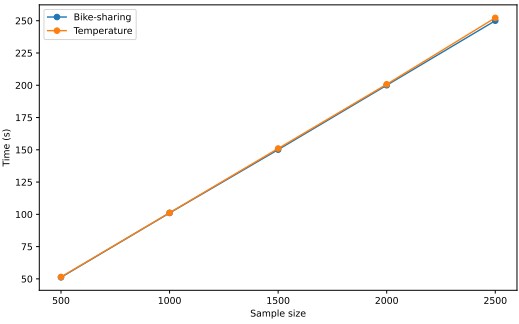

Figure 12: Runtime (seconds) of MoECP under different sample sizes. For instance, a 1000 sample size corresponds to training and calibration size equally to be 1000.

A.6    EXTENSION OF MOECP TO THE ONLINE SETTING

In online setting (Gibbs & Candes, 2021; Zaffran et al., 2022), we have $T_0$ observations $(X_1, Y_1), \cdots, (X_{T_0}, Y_{T_0})$ in $\mathbb{R}^d \times \mathbb{R}$. We aim to construct prediction intervals for $T_1$ subsequent observations $X_{T_0+1}, \cdots, X_{T_0+T_1}$ sequentially: at any prediction step $t \in [T_0 + 1, T_0 + T_1]$, $Y_{t-T_0}, \cdots, Y_{t-1}$ have been revealed. Therefore, the data $(X_{t-T_0}, Y_{t-T_0}), \cdots, (X_{t-1}, Y_{t-1})$ are available to construct the prediction interval for $Y_t$.

The online MoECP proceeds by maintaining and adaptively updating a target miscoverage level $\{\alpha_t\}$ based on recent performance and similarity-weighted errors. Let the nominal miscoverage be $\alpha \in (0, 1)$ and initialize $\alpha_{T_0+1} = \alpha$. For each prediction time $t = T_0 + 1, \ldots, T_0 + T_1$, form the MoE-based prediction set

$$\hat{\mathcal{C}}_{\alpha_t}^{\text{MoE}}(X_t) = \{y \in \mathbb{R} : S(X_t, y) \leq Q_{1-\alpha_t}\},$$

where $Q_{1-\alpha_t}$ denotes the $(1 - \alpha_t)$-quantile of the (weighted) conformity scores computed on the current calibration window. The miscoverage level is then updated by

$$\alpha_{t+1} = \alpha_t + \gamma \left( \alpha - \sum_{s=t-T_0+1}^{t} \bar{w}_s \mathbf{1}\{Y_s \notin \hat{\mathcal{C}}_{\alpha_s}^{\text{MoE}}(X_s)\} \right),$$

with step size $\gamma > 0$. The normalized cumulative weight $\bar{w}_s$ for $s \in [t - T_0 + 1, t]$ is defined as

$$\bar{w}_s = \frac{\sum_{i=t-T_0+1}^{s} w_i}{\sum_{i=t-T_0+1}^{t} w_i},$$

where each $w_i$ is computed analogously to Equation (4) using the current test covariate $X_t$. This update combines the recent empirical miscoverage frequency with similarity information between calibration and test points, making the online MoECP adaptive to recent errors while accounting for domain similarity. A thorough theoretical and empirical investigation of this online strategy is left for future work.

