# OpenReview forum: "Adaptive Conformal Prediction via Mixture-of-Experts Gating Similarity"
_ICLR.cc/2026/Conference — ICLR 2026 Poster_

### Official Review · Reviewer_c8Cp · 2025-10-15

**Soundness:** 3
**Presentation:** 3
**Contribution:** 3
**Rating:** 8
**Confidence:** 5

**Summary:**

The paper proposes a variant of the randomly localized conformal prediction (RLCP) introduced by Hore and Barber. The approach evaluates similarities between inputs in the latent space defined by the vector of probabilities output by the gating mechanism of a MoE model. The key idea is that inputs with similar statistics are also likely to be routed to the same experts if the gating mechanism is well trained.

**Strengths:**

The proposed approach is sound, retains statistical validity, and leverages advances in MoE architectures.

The experimental results, while limited, are sufficient to support the main claims of the paper.

**Weaknesses:**

The submission may not be sufficiently clear in the earlier sections about the relationship of this work with Hore-Barber. I think that this should be made clearer when presenting the contribution. As is, this is only discussed in Remark 5 on p. 5.

It is unclear a priori why the same temperature parameter tau is used in both (3) and (4).

Assumption 1 is not clear. The authors claim that the assumption holds for a variety of divergences, but they do not provide any supporting evidence for this.

There are some typos, such as "conformapl" on p. 1 and the missing space in "by(3)" in Algorithm 1.

**Questions:**

1) Why is the same temperature parameter tau used in both (3) and (4)?

2) Under what conditions is Assumption 1 verified?

---

> ### Author Response · Authors · 2025-11-21
> **Response to reviewer c8Cp (W1-W4)**
>
> >W1: The submission may not be sufficiently clear in the earlier sections about the relationship of this work with Hore-Barber. I think that this should be made clearer when presenting the contribution. As is, this is only discussed in Remark 5 on p. 5.
>
> To W1: Thank you for pointing this out! We agree that this relationship should be made clearer when presenting our contribution. Accordingly, the discussion of our work in relation to [1] has been revised in Line 59 as follows:
>
> *"... This work is related to randomly localized conformal prediction (RLCP) (Hore & Barber, 2025), where calibration points are chosen according to their proximity in the covariate space. By contrast, MoE-CP identifies relevant calibration points based on proximity in a *learned, label-aware latent regime space* defined by the gating probabilities of the mixture-of-experts. This makes the intervals not only valid, but also sharper, more flexible, and easier to interpret than other local weighted conformal methods, see Section 5 and Remark 5."*
>
> Reference:
>
> [1] Rohan Hore and Rina Foygel Barber. Conformal prediction with local weights: randomization enables robust guarantees. *Journal of the Royal Statistical Society Series B: Statistical Methodology*, 87(2):549–578, 2025.
>
> >W2: It is unclear a priori why the same temperature parameter tau is used in both (3) and (4).
>
> To W2: Thanks for going into the details, and apologies for any confusion. The theoretical justification for using the same temperature parameter $\tau$ is provided in the proof of Theorem 1. Here, we would like to highlight the key points:
>
> - In Equation (3), we sample $\tilde{L} \sim \text{Multinomial}(\tau, \pi(x_{n+1}))$.
>
> - After careful calculation, the weights used to compute the conformal quantile are
>
> $$\tilde{w}_i \propto\prod\_{k=1}^K[\pi\_k(x\_i)]^{\tilde{L}\_k}=\exp{\left(-\tau H(\tilde{\pi}(x\_{n+1}),\pi(x\_n))\right)}\propto \exp{\left(-\tau KL(\tilde{\pi}(x\_{n+1}),\pi(x\_n))\right)},$$
>
>   where $H(\cdot, \cdot)$ is the cross-entropy, and $KL(\cdot, \cdot)$ is the KL divergence. Note that the temperature $\tau$ remains present in the weights.
>
> This explains why the same temperature parameter $\tau$ is used in both Equations (3) and (4). We hope this explanation helps address your concern.
>
> >W3: Assumption 1 is not clear. The authors claim that the assumption holds for a variety of divergences, but they do not provide any supporting evidence for this.
>
> To W3: Thank you for the insightful comment, and we apologize for the lack of clarity. Assumption 1 holds whenever the divergence measure shares the same minimizer as the KL divergence. This condition, together with a detailed theoretical justification, is now provided in Line 766 of the revised manuscript. Moreover, the assumption is satisfied for many divergences when the calibration set is sufficiently large.
>
> The intuition is as follows. For a given $\tilde{\pi}:=\tilde{\pi}(x\_{n+1})$, if the calibration set contains a point whose gating vector $\pi\_j:=\pi(x\_j)$ equals (or arbitrarily closely approximates) $\tilde{\pi}$, then $KL(\tilde{\pi},\pi\_j)\approx 0$. Because $KL(\tilde{\pi},q)=0$ if and only if $q=\tilde{\pi}$. such a calibration point attains the minimizer of the KL divergence. For any divergence $D$ that also vanishes only when its arguments are equal, including Jeffreys divergence, the Hellinger distance, and the Euclidean distance, we similarly have $D(\tilde{\pi},\pi\_j)=0$. Therefore, the minimizer condition required by Assumption 1 is satisfied. These clarifications have been incorporated into Lines 283 and 756 of the revised manuscript. We hope this helps address your concern.
>
> >W4: There are some typos, such as "conformapl" on p. 1 and the missing space in "by(3)" in Algorithm 1.
>
> To W4: Thanks for the comments.  We have corrected the identified typos and have carefully reviewed the entire manuscript to address any additional errors. We appreciate your detailed feedback and thank you again for bringing this to our attention!

---

> ### Author Response · Authors · 2025-11-21
> **Response to reviewer c8Cp (Q1-Q2)**
>
> >Q1: Why is the same temperature parameter tau used in both (3) and (4)?
>
> To Q1: Thanks for the good comments. We would like to address your concern from the theoretical perspective on the construction of MoE-CP.
>
> - In Equation (3), we sample $\tilde{L} \sim \text{Multinomial}(\tau, \pi(x_{n+1}))$.
>
> - After careful calculation, the weights used to compute the conformal quantile are
>
> $$\tilde{w}_i \propto\prod\_{k=1}^K[\pi\_k(x\_i)]^{\tilde{L}\_k}=\exp{\left(-\tau H(\tilde{\pi}(x\_{n+1}),\pi(x\_n))\right)}\propto \exp{\left(-\tau KL(\tilde{\pi}(x\_{n+1}),\pi(x\_n))\right)},$$
> where $H(\cdot, \cdot)$ is the cross-entropy, and $KL(\cdot, \cdot)$ is the KL divergence. Note that the temperature $\tau$ remains present in the weights.
>
>  This is the reason why the same temperature parameter $\tau$ is used in both (3) and (4). We hope this interpretation addresses your concern.
>
> >Q2: Under what conditions is Assumption 1 verified?
>
> To Q2: Thanks for the helpful comments and apologies for the confusion. We have shown in the Appendix A.4.3 that Assumption 1 is verified if the divergence $D$ shares the same unique minimizer with the KL divergence, i.e., $i^*=\arg\min\_{i\in[n+1]}\tilde{w}\_{i,D}=\arg\min\_{i\in[n+1]}\tilde{w}\_{i,KL}$. To more clear clarification of this statement, we provide a formal proposition together with a detailed theoretical proof in the revised manuscript. These clarifications have been incorporated into Line 766 of the revised manuscript. We hope this explanation addresses your concerns.
>
> Thank you once again for your thoughtful feedback, and your questions have significantly improved the clarity and rigor of our paper!

---

> ### Comment · Reviewer_c8Cp · 2025-11-26
>
> Thank you for your clear replies. I am happy to confirm my initial positive assessment.

---

> > ### Author Response · Authors · 2025-11-26
> > **Official Comment by Authors**
> >
> > We sincerely appreciate your recognition and your valuable suggestions for improving the quality of our paper!

---

### Official Review · Reviewer_qsRG · 2025-10-30

**Soundness:** 2
**Presentation:** 2
**Contribution:** 2
**Rating:** 4
**Confidence:** 4

**Summary:**

This paper proposes MoE-CP, which uses MoE gating vectors as soft domain assignments to weight calibration residuals in conformal prediction. The method aims to produce adaptive prediction intervals for heterogeneous data without explicit domain labels, with theoretical guarantees for marginal validity and empirical validation showing tighter intervals than standard conformal methods.

**Strengths:**

Creative use of MoE gating vectors for conformal prediction weighting addresses heterogeneous data limitations.

Comprehensive analysis including marginal validity (Theorem 1), robustness to divergence choice (Theorem 2), and conditional coverage under mixture representation (Theorem 3).

**Weaknesses:**

Requires training an MoE model before conformal calibration, adding significant computational cost compared to standard conformal methods.

Multiple hyperparameters require careful tuning, and there is no detailed computational cost analysis compared to baseline methods.

Core assumption that MoE gating vectors meaningfully capture domain structure may not hold in practice, especially when true domains don't align with MoE's learned partitioning.

Limited discussion of how to validate whether MoE gating provides good domain separation.

MoE models are trained to minimize MSE loss, not to learn domain boundaries, creating fundamental misalignment between training objective and intended use.

Real datasets are low-dimensional, limiting generalizability to modern large-scale applications. Synthetic experiments only consider simple polynomial relationships with clear domain boundaries.

Randomization step in equation (3) appears somewhat ad-hoc without strong theoretical justification.

Exchangeability assumption may be violated in truly multi-domain data where domains have different distributions.

**Questions:**

The authors should address the concerns raised in the weakness section above.

---

> ### Author Response · Authors · 2025-11-21
> **Response to reviewer qsRG (W1-W2)**
>
> >W1: Requires training an MoE model before conformal calibration, adding significant computational cost compared to standard conformal methods.
>
> To W1: Thanks for the good comments. We agree that the dominant computational burden of MoE-CP is the training of the MoE model. We would like to emphasize three points:
>
> - **Offline cost.** Training the MoE is an offline cost shared by all downstream test queries; **the calibration weighting** step in MoE-CP is lightweight and fast in practice.
>
> - **MoE is a predictor.** The MoE is the base predictor used throughout the pipeline, so its training is not an extra, method-specific preprocessing step but the required model training step for any learned predictor. Compared with a single standard neural network, an MoE may incur additional compute because it contains multiple expert networks and a gating network; however, this investment yields a predictor that better captures latent multi-domain structure, which directly benefits interval efficiency and conditional adaptivity.
>
> - **Cost vs benifit.** While MoE training requires more computational resources than training a single model, the practical benefit is substantial: MoE-CP produces tighter, more adaptive prediction intervals while preserving coverage guarantees. In many applications where reliable, adaptive uncertainty quantification is important, this improved performance might justify the one-time training cost.
>
> We appreciate your thoughtful input and hope this clarification is satisfactory to you!
>
> >W2: Multiple hyperparameters require careful tuning, and there is no detailed computational cost analysis compared to baseline methods.
>
> To W2: Thanks for your helpful comments. We would like to respond to the two concerns separately.
>
> - **Hyperparameter selection.**  We agree that hyperparameter selection is an important consideration for any localized conformal method. In MoE-CP, the hyperparameters are the number of experts, the temperature parameter, and the divergence measure. We provided the hyperparameter selection in Appendix A.3, which is based on the ablation studies in Appendix A.4.3. In short, we found that: (1) a smaller number of experts $K$ (typically 2–3) is preferable when calibration/training data are limited, while larger $K$ can be used with abundant data; (2) the temperature in the range 100–300 gives a good trade-off between adaptivity and efficiency; (3) the KL divergence is the recommended divergence, while other divergences also work well.
>
> - **Computational complexity.**  We conducted experiments to measure the practical runtime on real datasets, averaging the results over 10 runs, using an AMD Ryzen 5 4600H with Radeon graphics. Table 1 reports the runtime for each method on the Bike-sharing and Temperature datasets, with training and calibration sizes set to 1500 and test size 1000, showing that the computational cost of MoE-CP is comparable to that of CHR and PCP. Table 2 presents the runtime for varying training/calibration sizes (test size fixed at 1000), and demonstrates that the runtime grows approximately linearly with sample size. We update the computational complexity analysis in Line 1227 of the revised manuscript.
>
> **Table 1: Runtime (seconds) for each method and dataset**
>
> | Methods      | CQR  | CHR    | SCP+CC | RLCP | PCP    | MoE-CP |
> |--------------|------:|-------:|-------:|-----:|-------:|-------:|
> | Bike-sharing | 3.3   | 151.7  | 27.0   | 5.1  | 134.7  | 153.2  |
> | Temperature  | 3.8   | 153.4  | 39.1   | 6.4  | 157.6  | 155.5  |
>
> **Table 2: Runtime (seconds) under different sample sizes (test size fixed at 1000)**
>
> | Sample Size  | 500   | 1000   | 1500   | 2000   | 2500   |
> |--------------|------:|-------:|-------:|-------:|-------:|
> | Bike-sharing | 51.2  | 101.0  | 150.1  | 200.0  | 250.0  |
> | Temperature  | 51.5  | 101.2  | 151.0  | 200.8  | 252.2  |

---

> ### Author Response · Authors · 2025-11-21
> **Response to reviewer qsRG (W3-W4)**
>
> >W3: Core assumption that MoE gating vectors meaningfully capture domain structure may not hold in practice, especially when true domains don't align with MoE's learned partitioning.
>
> To W3: Thank you for this thoughtful question! This is a central and valid concern since MoE is very popular in current real world model training (LLM,CV, MLP). We would like to emphasize two points:
>
> - **Safety.** The marginal validity guarantee of MoE-CP (Theorem~1) holds for any fixed set of weights produced by the gating network; it does not require that the gating perfectly recover the true domain structure. As long as the calibration and test points are exchangeable and we use the randomized weighting scheme specified in the theorem, MoE-CP achieves exact marginal coverage. In the extreme case where the gating is uninformative (e.g., nearly constant), MoE-CP essentially behaves like a global weighted CP method and remains valid, albeit with little or no conditional adaptivity.
>
> - **Adaptivity.** Conditional adaptivity gains do depend on gating quality (Theorem 3). Please note that conditional coverage guarantee is hard to satisfy. It has been shown that the conditional coverage can not be achieved in a distribution-free setting, that is, without distributional assumptions; otherwise, the resulting prediction intervals would be uninformative [1,2]. Here, we mention that if the MoE model meaningfully captures the domain structure, the conditional coverage guarantee of MoE-CP can approximately hold. Empirically from Figure 3 and 4, what we observe does imply that the MoE model captures well, since the conditional coverage is approximately held.
>
> We hope that the above interpretation can ease your doubts.
>
> Reference:
>
> [1] Lei, J., and Wasserman, L. (2014). Distribution-free prediction bands for non-parametric regression. Journal of the Royal Statistical Society Series B: Statistical Methodology, 76(1), 71-96.
>
> [2] Foygel Barber, et al. (2021). The limits of distribution-free conditional predictive inference. Information and Inference: A Journal of the IMA, 10(2), 455-482.
>
> >W4: Limited discussion of how to validate whether MoE gating provides good domain separation.
>
> To W4: Thank you for the good comments. We agree that whether the learned gating $\pi(\cdot)$ meaningfully reflects latent domains is a key practical question because it determines how much conditional adaptivity MoE-CP can realize. Our focus here is prediction intervals adaptive to heterogeneous and latent multi-domain data. The conditional adaptivity of MoE-CP does depend on whether $\pi(\cdot)$ correlates with predictive heterogeneity. Therefore, our focus is not “does the MoE cover the true domains?”, but rather “does the learned $\pi(\cdot)$ provide useful signal for improving conditional coverage and interval length?”. This is an empirical question, which we evaluate by measuring interval length and conditional coverage of the prediction intervals.
>
> Furthermore, prior literature shows that MoE gating often provides effective domain separation. For example, [1] demonstrates that the router learns cluster-center features and partitions the input into simpler subproblems, while [2] empirically shows that a trainable router on cluster-structured data recovers the underlying clusters. We will include these MoE discussions in Related Work to further justify the merits and insights of using MoE.
>
> References
>
> [1] Chen, Z., et al. (2022). *Towards understanding the mixture-of-experts layer in deep learning.* Advances in Neural Information Processing Systems, 35, 23049–23062.
>
> [2]Dikkala, N., et al. (2023). *On the benefits of learning to route in mixture-of-experts models.* Proceedings of the 2023 Conference on Empirical Methods in Natural Language Processing, 9376–9396.

---

> ### Author Response · Authors · 2025-11-21
> **Response to reviewer qsRG (W5-W7)**
>
> >W5: MoE models are trained to minimize MSE loss, not to learn domain boundaries, creating fundamental misalignment between training objective and intended use.
>
> To W5: Thank you for this insightful comment. We agree that the MoE is trained to minimize predictive loss (MSE), not to directly recover ground-truth domain labels. However, our use of the gating vectors is consistent with this objective and does not require learning sharp domain boundaries.
>
> Conceptually, the MoE behaves more like a regression mixture model than a hard classifier such as an SVM. An SVM is trained to learn a clear decision boundary, whereas an MoE defines a set of experts and a gating distribution $\pi(x)$ over them. This is analogous to a Gaussian mixture model where EM is used to learn mixture probabilities $\pi$; here, an MLP instead learns a flexible, input-dependent probability vector $\pi(x)$ that reflects soft responsibilities for latent predictive regimes.
>
> In MoE-CP, we only need $\pi(x)$ to serve as a **soft domain assignment**: calibration points with gating vectors similar to that of the test point are inferred to share a similar conditional distribution and are given higher weights when computing the local $(1-\alpha)$ threshold. We never require explicit domain labels or exact domain boundaries. When the gating is informative, this yields conditional adaptivity; when it is not, MoE-CP smoothly falls back toward global behavior while preserving marginal coverage. We sincerely hope our clarifications resolve your concerns!
>
> >W6: Real datasets are low-dimensional, limiting generalizability to modern large-scale applications. Synthetic experiments only consider simple polynomial relationships with clear domain boundaries.
>
> To W6: Thanks for the good comments, and apologies for the limited experimental results in our study. In our work, our primary focus is on developing and validating the MoE-based conformal prediction methodology itself (rather than specializing in MoE modeling). The synthetic experiments with polynomial data are intended to provide a clear visualization and conceptual understanding. As Reviewer c8Cp noted, ``The experimental results, while limited, are sufficient to support the main claims of the paper". Indeed, MoE has demonstrated strong performance in complex, high-dimensional data scenarios and modern large-scale applications, e.g.,  pedestrian classification [1], CV [2], and LLM [3]. However, these are somewhat beyond our current capabilities. We would like to emphasize that our proposed method is designed as a versatile tool that can be easily implemented by practitioners who work with MoE models.
>
> Reference:
>
> [1] Enzweiler, M., and Gavrila, D. M. (2011). A multilevel mixture-of-experts framework for pedestrian classification. IEEE Transactions on Image Processing, 20(10), 2967-2979.
>
> [2] Lin, B., et al. (2024). Moe-llava: Mixture of experts for large vision-language models. arXiv preprint arXiv:2401.15947.
>
> [3] Cai, W., et al. (2025). A survey on mixture of experts in large language models. IEEE Transactions on Knowledge and Data Engineering.
>
> >W7: Randomization step in equation (3) appears somewhat ad-hoc without strong theoretical justification.
>
> To W7: Thanks for the comments. The multinomial randomization step in Equation (3) has theoretical justification; please refer to the proof of Theorem 1, where this randomization is essential. Here, we provide the key points:
>
> - At first, we draw a sample size $\tau$ from a multinomial distribution, i.e., $\tilde{L} \sim \text{Multinomial}(\tau, \pi(X_{n+1}))$ where $\pi(X_{n+1})$ is the gating probability of $X_{n+1}$.
>
> - Denote $Z_i = (X_i, Y_i)$ for $i \in [n+1]$, and let $z_i$ be the realization of $Z_i$. By exchangeability,
>   $$\mathbb{P}\lbrace Z_{n+1} = z_i\\rbrace = \frac{1}{n+1}, \quad i \in [n+1].$$
>
> - Using Bayes' theorem to obtain the posterior distribution as
>   $$\mathbb{P}\lbrace Z_{n+1} = z_i \mid \tilde{\pi}(X_{n+1})\rbrace \propto \prod_{k=1}^K [\pi_k(x_i)]^{\tilde{L}_k},$$
> which corresponds to the likelihood function $\text{Multinomial}(\tau, \pi(x_i))$ evaluated at $\tilde{L}$, and $\pi(x_i)$ is the gating probability at $x_i$.
>
> Please note that this multinomial sampling is crucial for the construction of MoE-CP. We hope this explanation helps address your concern!

---

> ### Author Response · Authors · 2025-11-21
> **Response to reviewer qsRG (W8)**
>
> >W8: Exchangeability assumption may be violated in truly multi-domain data where domains have different distributions.
>
> To W8: In our work we assume the data are i.i.d. draws from some overall distribution $P$. Concretely, we model $P$ as a mixture that may contain $K$ latent domains, but we treat all observed samples as draws from the same distribution $P$. Under this modeling choice the usual exchangeability assumption (i.i.d. sampling from $P$) holds, and it is this assumption that underlies the finite-sample marginal validity in Theorem 1.
>
> We agree that there is another useful formalism, called hierarchical exchangeability, in which (1) different domains are exchangeable as groups and (2) observations within each domain are exchangeable. That framework is appropriate when domain/group labels are known [1, 2, 3]. In our setting the domain labels are latent and not observed; consequently the hierarchical exchangeability framework does not directly apply without first identifying or observing group membership. We acknowledge that how MoE-CP could be extended to settings with known domain labels or to non-exchangeable regimes deserves future work. We have added this insightful point to the conclusion in Sec 6 in Line 481.
>
> Thank you for your valuable feedback and for taking the time to engage with our work. We hope this explanation is satisfactory to you!
>
> Reference:
>
> [1] Lee, Y., Barber, R. F., and Willett, R. (2023). Distribution-free inference with hierarchical data. arXiv preprint arXiv:2306.06342.
>
> [2] Dunn, R., Wasserman, L., and Ramdas, A. (2023). Distribution-free prediction sets for two-layer hierarchical models. Journal of the American Statistical Association, 118(544), 2491-2502.
>
> [2] Duchi, J. C., et al. (2025). Predictive inference in multi-environment scenarios. Statistical Science, 40(3), 392-416.

---

> ### Author Response · Authors · 2025-11-28
> **Looking forward to your feedback**
>
> Dear Reviewer,
>
> We hope this message finds you well. With the discussion period ending in fewer than five days, we want to ensure we have addressed all your concerns satisfactorily. If you have any additional comments or suggestions, please let us know. Your insights are invaluable, and we would be happy to address any remaining points to further improve our work.
>
> Thank you again for your time and effort in reviewing our paper!
>
> Best wishes,
>
> The Authors

---

### Official Review · Reviewer_ZRke · 2025-10-31

**Soundness:** 2
**Presentation:** 3
**Contribution:** 2
**Rating:** 4
**Confidence:** 3

**Summary:**

The paper proposes Mixture-of-Experts Conformal Prediction, which integrates conformal prediction with mixture-of-experts models. It assigns higher weights to calibration samples whose gating vectors are more similar to the test point’s, thereby adapting prediction intervals to latent domain structures.

**Strengths:**

1- The paper introduces an interesting and well-motivated idea that combines Mixture-of-Experts (MoE) models with conformal prediction

2- The paper provides a theoretical analysis, proving marginal validity and approximate conditional coverage

**Weaknesses:**

Please check the questions!

**Questions:**

1- The authors should clarify the precise meaning of $\mu(x)$ and $\pi(x)$. While both are formally defined, their conceptual interpretation and how they are learned from data remain vague.

2- It is unclear how the number of experts and the temperature parameter are selected in practice. Please explain how you chose these parameters.

3- Since the approach involves computing similarity-based weights for each test point, the computational complexity may be substantial for large calibration sets. It would be helpful if the authors could provide a computational complexity analysis and discuss practical runtime implications.

4- There are existing conformal prediction approaches that incorporate expert advice [1, 2]; the authors are encouraged to provide a comparison with these methods, either conceptually or empirically

5-The impact of the randomization step (via multinomial sampling in Eq. 3) on interval variability and stability is not analyzed. Please provide an analysis to show its impact

[1] Hajihashemi, E. and Shen, Y., 2024. Multi-model ensemble conformal prediction in dynamic environments. Advances in Neural Information Processing Systems, 37, pp.118678-118700.
[2] Gibbs, I. and Candès, E.J., 2024. Conformal inference for online prediction with arbitrary distribution shifts. Journal of Machine Learning Research, 25(162), pp.1-36.

---

> ### Author Response · Authors · 2025-11-21
> **Response to reviewer ZRke (Q1-Q2)**
>
> > Q1: The authors should clarify the precise meaning of $\mu(x)$ and $\pi(x)$. While both are formally defined, their conceptual interpretation and how they are learned from data remain vague.
>
> To Q1: We apologize for the lack of clarity and thank the reviewer for pointing this out. We are happy to clarify the conceptual meaning of $\mu(x)$ and $\pi(x)$ and how they are learned from data.
>
> - **Interpretation of $\mu(x)$.**  For each expert $k \in \{1,\dots,K\}$, $\mu_k(x)$ denotes the prediction produced by expert $k$ for input $x$. The overall MoE prediction is the mixture $\mu(x)=\sum_{k=1}^K \pi_k(x)\mu_k(x)$, i.e., a weighted average of expert outputs. Conceptually, $\mu(x)$ is the model’s point predictor for $Y$ given $X=x$, and it is this predictor that we use to form conformal scores.
>
> - **Interpretation of $\pi(x)$.**  The vector $\pi(x)=(\pi_1(x),\dots,\pi_K(x))$ is the output of the gating network, with $\pi_k(x)\ge 0$ and $\sum_{k=1}^K \pi_k(x)=1$. It can be viewed as a probability distribution over latent experts or domains, i.e., $\pi_k(x)\approx P(Z=k\mid X=x)$ for a latent variable $Z$ indicating which expert is responsible for predicting $Y$ at $x$. Intuitively, $\pi_k(x)$ expresses the model's learned belief about which expert(s) are responsible for predicting at $x$. It is a soft, task-relevant assignment of $x$ to latent predictive regimes.
>
> - **How they are learned.**  Both the experts $\lbrace\mu_k\rbrace_{k=1}^K$ and the gating network that outputs $\pi(\cdot)$ are parameterized (e.g., by neural networks) and trained jointly by minimizing a standard predictive loss (e.g., MSE) on the training data. Gradients flow through both the experts and the gate, so the gating function learns to route different inputs to different experts in a way that reduces prediction error. As a result, experts specialize and $\pi(x)$ becomes a semantically meaningful, low-dimensional representation of predictive regime membership.
>
> >Q2: It is unclear how the number of experts and the temperature parameter are selected in practice. Please explain how you chose these parameters.
>
> To Q2: Thanks for the comments. We provide practical guidance in *Appendix A.3*, which is based on the ablation study in *Appendix A.4.3*. In the synthetic data experiments, we set $K=3$ and $\tau=150$; in the real-data experiments, we set $K=2$ and $\tau=100$. Here, we summarize and clarify how the number of experts and the temperature are selected.
>
> - **Choosing the number of experts K.** If calibration/training sample size is modest (for example $n\leq 1500$), use a small $K$, e.g., $K=2,3$ to avoid fragmenting calibration data across too many experts. This preserves effective sample size per regime and produces more efficient intervals. With abundant data and clearly multimodal regimes, larger $K$ can capture finer-grained structure. One can also fit an over-parameterized MoE and use sparsity/regularization to find an effective $K$.
>
> - **Choosing the temperature parameter $\tau$.** $\tau$ trades off adaptivity vs effective sample size: small $\tau$ leads to more randomization and more uniform weights, large $\tau$ leads to weights that concentrate on a few calibration points. The choice of $\tau$ is data-driven and can be referred to [1]. We conduct several ablation studies in *Appendix A.4.3* and found $\tau\in[100,300]$ to be a robust default.
>
> Reference:
>
> [1] Zhang, Y., and Candès, E. J. (2024). Posterior conformal prediction. arXiv preprint arXiv:2409.19712.

---

> ### Author Response · Authors · 2025-11-21
> **Response to reviewer ZRke (Q3-Q4)**
>
> >Q3: Since the approach involves computing similarity-based weights for each test point, the computational complexity may be substantial for large calibration sets. It would be helpful if the authors could provide a computational complexity analysis and discuss practical runtime implications.
>
> To Q3: Thank you for this helpful comment! We agree that a clear complexity discussion will improve the paper. We clarify below that the dominant computational cost lies in training the MoE model, while the calibration step of MoE-CP is lightweight.
>
> - **Training cost (dominant).**  Training the MoE involves forward/backward passes through the gating network and all experts. If $m$ is the training sample size, $K$ the number of experts, and $E$ the number of epochs, the overall training cost is on the order of $O\big(E m (C_{\text{gate}} + K C_{\text{exp}})\big)$, where $C_{\text{gate}}$ and $C_{\text{exp}}$ denote one forward-backward cost of the gate and a single expert, respectively. This is a one-time offline cost and represents the main computational burden of the method. In our experiments, training is performed once per dataset and is easily handled.
>
> - **Calibration cost (lightweight).**  Once the MoE is trained, we precompute the gating vectors $\pi(X_i), i\in[n]$, of the calibration set. For each test point, MoE-CP computes (1) one forward pass of the gating network, and (2) pairwise divergences between the test gating vector and the $n$ calibration gating vectors. This results in an $O(n)$ cost per test point, which is extremely fast in practice. The computational complexity of the calibration stage is analogous to randomly localized conformal prediction [1].
>
> We conducted experiments to measure the practical runtime on real datasets, averaging the results over 10 runs, using an AMD Ryzen 5 4600H with Radeon graphics. Table 1 reports the runtime for each method on the Bike-sharing and Temperature datasets, with training and calibration sizes set to 1500 and test size 1000, showing that the computational cost of MoE-CP is comparable to that of CHR and PCP. Table 2 presents the runtime for varying training/calibration sizes (test size fixed at 1000), and demonstrates that the runtime grows approximately linearly with sample size. We update the computational complexity analysis in Line 1227 of the revised manuscript.
>
> **Table 1: Runtime (seconds) for each method and dataset**
>
> | Methods      | CQR  | CHR    | SCP+CC | RLCP | PCP    | MoE-CP |
> |--------------|------:|-------:|-------:|-----:|-------:|-------:|
> | Bike-sharing | 3.3   | 151.7  | 27.0   | 5.1  | 134.7  | 153.2  |
> | Temperature  | 3.8   | 153.4  | 39.1   | 6.4  | 157.6  | 155.5  |
>
> **Table 2: Runtime (seconds) under different sample sizes (test size fixed at 1000)**
>
> | Sample Size  | 500   | 1000   | 1500   | 2000   | 2500   |
> |--------------|------:|-------:|-------:|-------:|-------:|
> | Bike-sharing | 51.2  | 101.0  | 150.1  | 200.0  | 250.0  |
> | Temperature  | 51.5  | 101.2  | 151.0  | 200.8  | 252.2  |
>
> Reference
>
> [1] Rohan Hore and Rina Foygel Barber. *Conformal prediction with local weights: randomization enables robust guarantees.* Journal of the Royal Statistical Society Series B: Statistical Methodology, 87(2):549–578, 2025.
>
> >Q4: There are existing conformal prediction approaches that incorporate expert advice [1, 2]; the authors are encouraged to provide a comparison with these methods, either conceptually or empirically.
>
> To Q4: Thank you for pointing out these related directions. We would like to provode the distinction between our and their works. [1, 2] address online settings where the data stream and/or model pool evolve over time, and methods must adapt to distribution shift as it arrives. Their methods are designed for sequential prediction and robustness to arbitrary or time-varying shifts. By contrast, MoE-CP is developed for the offline setting: we train a single Mixture-of-Experts predictor on available training data and then use its learned gating vectors to produce locality-aware conformal calibration for test queries. Our focus is on exploiting latent domain structure in (static) datasets rather than online adaptation.
>
> Reference:
>
> [1] Hajihashemi, E. and Shen, Y., 2024. Multi-model ensemble conformal prediction in dynamic environments. Advances in Neural Information Processing Systems, 37, pp.118678-118700.
>
> [2] Gibbs, I. and Candès, E.J., 2024. Conformal inference for online prediction with arbitrary distribution shifts. Journal of Machine Learning Research, 25(162), pp.1-36.

---

> ### Author Response · Authors · 2025-11-21
> **Response to reviewer ZRke (Q5)**
>
> >Q5: The impact of the randomization step (via multinomial sampling in Eq. 3) on interval variability and stability is not analyzed. Please provide an analysis to show its impact.
>
> To Q5: Thanks for raising this important point! We consider the following metric for interval variability and stability, as introduced in Appendix C.1 of [1]:
> $$
> D(\tau)=\mathbb{E}\left[
> \frac{\mathrm{MAD}\bigl(\lambda(C(X_{n+1}))\bigr)}
>      {\mathrm{Median}\bigl(\lambda(C(X_{n+1}))\bigr)}
> \right],
> $$
> where $\lambda(\cdot)$ denotes the Lebesgue measure of a prediction interval, and  $\mathrm{MAD}(X)=\mathrm{Median}(|X-\mathrm{Median}(X)|)$.  The only source of randomness is the multinomial draw. In other words, $D(\tau)$ quantifies the variability of the MoE-CP interval width. Table 1 reports the results on interval variability for two real datasets, indicating that the metric is robust to the choice of $\tau$. This analysis is included with a detailed description in Line 1108 of the revised manuscript.
>
> Thank you once again for your effort and insightful questions about our work; they have greatly helped us improve our paper! We hope these responses help address your concerns.
>
> ---
>
> **Table 1: The effects of randomization on MoE-CP through the estimated deviation $D(\tau)$.**
>
> | **τ**        | **50**  | **100** | **150** | **200** | **250** | **300** |
> |--------------|--------:|--------:|--------:|--------:|--------:|--------:|
> | Bike-sharing | 0.0314  | 0.0384  | 0.0410  | 0.0283  | 0.0254  | 0.0303  |
> | Temperature  | 0.0169  | 0.0185  | 0.0229  | 0.0201  | 0.0259  | 0.0275  |
>
> ---
>
> Reference
>
> [1] Hore, R., and Barber, R. F. (2025). *Conformal prediction with local weights: randomization enables robust guarantees*. Journal of the Royal Statistical Society Series B: Statistical Methodology, 87(2), 549–578.

---

> ### Author Response · Authors · 2025-11-28
> **Looking forward to your feedback**
>
> Dear Reviewer,
>
> We hope this message finds you well. With the discussion period ending in fewer than five days, we want to ensure we have addressed all your concerns satisfactorily. If you have any additional comments or suggestions, please let us know. Your insights are invaluable, and we would be happy to address any remaining points to further improve our work.
>
> Thank you again for your time and effort in reviewing our paper!
>
> Best wishes,
>
> The Authors

---

### Official Review · Reviewer_TKSb · 2025-11-02

**Soundness:** 2
**Presentation:** 2
**Contribution:** 2
**Rating:** 4
**Confidence:** 4

**Summary:**

The authors introduce Mixture-of-Experts Conformal Prediction (MoE-CP), a flexible method that uses the gating probability vectors of Mixture-of-Experts (MoE) models to estimate the similarity between test sample and calibration samples, thereby achieve adaptive conformal prediction.

**Strengths:**

1. Sufficient discussion about background and clear introductive figure.

2. Extensive ablation study on hyperparameter selection of the proposed method.

**Weaknesses:**

1. This work basically follows the logics of prior research: compute the similarity between calibration samples and a give test sample to estimate local conformal score density, which is used to output a local 1-alpha threshold for prediction sets. Thereby, the overall contribution is incremental.

2. The gate model typically plays a role of PCA to facilitate the density estimation. In other words, the number of expert models K must be sufficiently smaller than feature dimensions to be functional, which limits the application of the proposed method.

3. Density-estimation-based localized CP , such as [1,2], are sensitive to hyperparameter selection. The authors only mention which base predictive models are used, without discussing if their hyperparameters are tuned carefully. Hence, the experiement results may be unfair. 4. Recently, generative models for adaptiveness progressed, such as [3,4], which are missed in the work.

[1] Leying Guan. Localized conformal prediction: A generalized inference framework for conformal prediction. Biometrika, 110(1):33–50, 2023.
[2] Rohan Hore and Rina Foygel Barber. Conformal prediction with local weights: randomization enables robust guarantees. Journal of the Royal Statistical Society Series B: Statistical Methodology, 87(2):549–578, 2025.
[3] Colombo, Nicolo. "Normalizing flows for conformal regression." arXiv preprint arXiv:2406.03346 (2024).
[4] Fang, Zhenhan, Aixin Tan, and Jian Huang. "CONTRA: Conformal prediction region via normalizing flow transformation." The Thirteenth International Conference on Learning Representations. 2025.

**Questions:**

1. What about extending your idea to classification? What challenge will you face?

2. What is the difference between the similarity weights from gate model and a density estimator?

---

> ### Author Response · Authors · 2025-11-21
> **Response to reviewer TKSb (W1-W2)**
>
> > W1: This work basically follows the logics of prior research: compute the similarity between calibration samples and a give test sample to estimate local conformal score density, which is used to output a local 1-alpha threshold for prediction sets. Thereby, the overall contribution is incremental.
>
> To W1: Thank you for your valuable feedback and for taking the time to engage with our work. We appreciate your insightful comments and suggestions. We agree that all the weighted conformal methods fit the general paradigm originating from [3], and our method is within this scope. Our contribution is to show that, in the context of MoE models, using the **learned gating probabilities** as the similarity notion leads to both new theory and substantially improved practical performance beyond prior work such as [1, 2].
>
> - **New similarity based on MoE gating vectors.** Prior localized CP methods [1,2] typically define locality in covariate space via distance/clustering in $X$. In contrast, MoE-CP uses locality in **gating probability space** that reflects joint heterogeneity in $(X, Y)$ and latent domains; it is better suited to multi-domain/distribution-shift scenarios via **soft domain assignment**. This yields similarity that is (a) **target-aware**, because the gating network is trained to capture predictive regimes relevant to the response $y$, whereas covariate distances do not encode this label-relevant structure [1,2]; (b) **low-dimensional and semantically meaningful** even when the input is high-dimensional, since each gating coordinate corresponds to an expert/latent regime; and (c) **interpretable**, as proximity in gating space directly reflects sharing the same mixture component. As we show in Figure 4, this leads to tighter prediction sets and better conditional adaptivity in heterogeneous settings where covariate-based locality alone is insufficient.
>
> - **Theoretical novelty for divergence-based gating similarities.** We provide new guarantees tailored to MoE-based similarity weights. In particular, we establish exact marginal validity for KL/cross-entropy-type weights (Theorem 1), asymptotic robustness for a broad family of divergence-based weightings (Theorem 2), and conditional coverage under a mixture-of-experts representation (Theorem 3). To the best of our knowledge, we are the first to use MoE gating probabilities as the similarity measure in conformal prediction and to rigorously characterize coverage properties in this setting.
>
> Reference:
>
> [1] Leying Guan. Localized conformal prediction: A generalized inference framework for conformal prediction. Biometrika, 110(1):33–50, 2023.
>
> [2] Rohan Hore and Rina Foygel Barber. Conformal prediction with local weights: randomization enables robust guarantees. Journal of the Royal Statistical Society Series B: Statistical Methodology, 87(2):549–578, 2025.
>
> [3] Barber, R. F.,  et al. (2023). Conformal prediction beyond exchangeability. The Annals of Statistics, 51(2), 816–845.
>
> > W2: The gate model typically plays a role of PCA to facilitate the density estimation. In other words, the number of expert models $K$ must be sufficiently smaller than feature dimensions to be functional, which limits the application of the proposed method.
>
> To W2: Thank you for the careful reading and good comments. We would like to clarify two points, namely the role of the MoE gating network and the interpretation of the number of experts $K$.
>
> - **Gating is not PCA.**  The MoE gating network is typically a discriminatively trained (e.g., MLP) function whose softmax output $\pi(X)=\text{softmax}(\ell(X))$ gives a probability over experts for each input. The gating and expert networks are trained jointly to optimize the predictive objective, so $\pi(X)$ encodes which expert(s) are responsible for predicting $Y$ at $X$. This representation is **task-aware** and directly tied to the conditional distribution of $Y$ given $X$, rather than capturing raw input variance as PCA does. In particular, $\pi(X)$ often reflects latent domains or regimes. Using similarity in the space of gating probabilities is therefore fundamentally different from applying PCA to $X$ and performing density estimation on a low-dimensional projection: it uses a label-informed, domain-aware representation instead of an unsupervised projection of the covariates.
>
> - **$K$ smaller than feature dimension.**  We agree that in practice the number of experts $K$ is typically smaller than the feature dimension, but we do not view this as a limitation of the method. Conceptually, $K$ controls the **number of latent regimes**, not the effective dimension of $X$: we choose $K$ to match the heterogeneity we want to capture, e.g., the number of domains, rather than to approximate the covariate space itself. Our conformal guarantees hold for any fixed $K$; increasing $K$ trades off between model complexity and data requirements but is not constrained by the ambient feature dimension.

---

> ### Author Response · Authors · 2025-11-21
> **Response to reviewer TKSb (W3-W4, Q1-Q2)**
>
> >W3: Density-estimation-based localized CP, such as [1,2], are sensitive to hyperparameter selection. The authors only mention which base predictive models are used, without discussing if their hyperparameters are tuned carefully. Hence, the experiement results may be unfair.
>
> To W3: In our experiments, we include RLCP [1] as a representative density-estimation–based localized CP method. For RLCP, the base predictive model is implemented using `sklearn.ensemble.RandomForestRegressor`. Unless otherwise specified, we adopt the standard `scikit-learn` default hyperparameters for the random forest and set the kernel bandwidth following the recommendation in [1], namely choosing the bandwidth such that the effective sample size equals 100. Thus, RLCP is configured according to the settings suggested in the original paper rather than being under-tuned. In Line 876 of the revised manuscript, we have added the RLCP hyperparameter choices for clarification.
>
> Reference
>
> [1] Rohan Hore and Rina Foygel Barber. Conformal prediction with local weights: randomization enables robust guarantees. Journal of the Royal Statistical Society Series B: Statistical Methodology, 87(2):549–578, 2025.
>
> > W4: Recently, generative models for adaptiveness progressed, such as [3,4], which are missed in the work.
>
> To W4: Thank you for raising this important direction! References [3] and [4] propose new conformal scores based on normalizing flows: [3] targets one-dimensional outputs, whereas [4] addresses multivariate outputs. By contrast, our work investigates how to assign weights within conformal scores to improve adaptivity. We discuss these two papers in Line 465 of the revised manuscript.
>
> >Q1: What about extending your idea to classification? What challenge will you face?
>
> To Q1: Thank you for the constructive comments. You raise an important and interesting question! For classification, we can similarly train an MoE classifier of the form $f_Y( X)=\sum_{k=1}^K\pi_k(X)f_{k,Y}(X)$, where $f_Y(X)$ is the MoE's classifier's predictive probability of categorizing the object with feature $X$ as label $Y$, and $\pi(X)$ is the gating probability.
>
> A natural extension to the classification setting is to replace the regression conformity score $|Y-\hat{\mu}(X)|$ with the classification conformity score $1-f_Y(X)$ [1], using the same reasoning as in the regression case. In addition, one can define an MoE-based conformity score to construct adaptive prediction sets. For example, following the idea in [2], we may design an adaptive conformity score that incorporates the gating information. We acknowledge that MoE-based conformal prediction for classification is an important direction that needs further investigation, with many open questions remaining.
>
> Reference:
>
> [1] Sadinle, M., Lei, J., and Wasserman, L. (2019). Least ambiguous set-valued classifiers with bounded error levels. Journal of the American Statistical Association, 114(525), 223-234.
>
> [2] Romano, Y., Sesia, M., and Candes, E. (2020). Classification with valid and adaptive coverage. Advances in neural information processing systems, 33, 3581-3591.
>
> >Q2: What is the difference between the similarity weights from gate model and a density estimator?
>
> To Q2: Thanks for your good comments! At a high level, density-estimation-based similarity weights measure proximity in the **raw covariate space** $X$, whereas our gating-based similarity measures proximity in a **learned, label-aware latent regime space** defined by the MoE gating probabilities.
>
> - **Density-based similarity.**  Density-based similarity relies on a density estimator to evaluate the distance between the covariates of a calibration point $X_i$ and the test point $X_{n+1}$, typically using metrics such as $||X_i - X_{n+1}||_2$. Calibration points that are closer in Euclidean distance receive higher weights when computing the local $1-\alpha$ threshold for the prediction set.
>
> - **Gating-based similarity.**  Gating-based similarity uses the learned MoE gating probabilities $\pi(X)$ to capture “which predictive expert is responsible for this input”. This representation is supervised and prediction-aware. Calibration points with gating vectors similar to that of the test point are treated as arising from a similar data-generating regime and therefore receive higher weights in computing the local $1-\alpha$ threshold.
>
> We hope this explanation clarifies the distinction between our method and density-based localized CP. Please also refer to our response to reviewer c8Cp in Weaknesses 1, where we discuss the relationship between our method and density-based localization. We appreciate your thoughtful input and hope this explanation is satisfactory to you!

---

> ### Author Response · Authors · 2025-11-28
> **Looking forward to your feedback**
>
> Dear Reviewer,
>
> We hope this message finds you well. With the discussion period ending in fewer than five days, we want to ensure we have addressed all your concerns satisfactorily. If you have any additional comments or suggestions, please let us know. Your insights are invaluable, and we would be happy to address any remaining points to further improve our work.
>
> Thank you again for your time and effort in reviewing our paper!
>
> Best wishes,
>
> The Authors

---

### Author Response · Authors · 2025-12-01
**Overview Response to Reviewers and Area Chairs**

We thank the reviewers and area chairs for their careful reading and constructive comments. Below, we (1) clarify our main contributions, (2) summarize and address the key concerns raised by multiple reviewers, and (3) list the substantive manuscript revisions we have incorporated.

---
## **Contributions.**
Briefly, our main contribution is to show that, for mixture-of-experts (MoE) models, using the learned gating probabilities as a similarity measure yields **both new theoretical insights and substantial practical improvements** over prior approaches. The gating-based similarity is **target-aware, low-dimensional, semantically meaningful, and interpretable**; these properties produce tighter prediction sets and better conditional adaptivity in heterogeneous settings. On the theory side, we introduce fundamentally new results for gating-based similarity weighting: **exact marginal validity for KL/cross-entropy–type weights, asymptotic robustness for a broad family of divergence-based weightings, and conditional coverage guarantees under a mixture-of-experts representation**. To the best of our knowledge, we are the first to (1) use MoE gating probabilities as a similarity measure in conformal prediction and (2) rigorously characterize the coverage properties that follow. **Reviewer c8Cp (confidence level 5) recognized and endorsed these contributions and reaffirmed this view after our initial reply.**

---

## **Key concerns raised by reviewers and our responses.**

- **The MoE model.** Several reviewers sought clarification about MoE structure and training. Reviewer ZRke appeared unsure about what an MoE is and how its components are learned; in response, we provide a concise conceptual description of the expert outputs μ(x) and the gating probabilities π(x) and explain the learning procedure. Reviewer TKSb compared the gating to PCA; we clarify that gating is not an unsupervised projection but a label-informed, domain-aware representation learned jointly with the experts.

- **Gating-based similarity weighting.** Reviewers expressed multiple concerns here. Reviewer TKSb worried that the method would be limited when the number of experts K is smaller than the covariate dimension. We explain why this is not a practical limitation: the gating provides a low-dimensional, label-aware encoding that can be more informative than raw high-dimensional covariates. Reviewer qsRG suggested the MoE must learn sharp domain boundaries; we show that sharp boundaries are unnecessary — the gating only needs to create a soft domain assignment that reflects regime structure. Reviewer qsRG also raised questions about the randomization step used in weighting; in response, we provide a clear theoretical justification for the randomization step.

- **Experiments and computational analysis.** Reviewers ZRke and qsRG asked for computational-complexity analysis; we have added this analysis to the revised manuscript. Reviewer ZRke also asked about the effect of randomization on interval variability and stability; we analyze variability and stability in the revision and report the results.

We regret that Reviewers TKSb, ZRke, and qsRG did not submit follow-up comments after our initial reply. **We believe their concerns were minor and largely the result of misunderstandings about MoE; these misunderstandings are now addressed by our clarifications.** We hope the revisions and added explanations resolve the issues they raised.

---

## **Manuscript revisions.**

- We expanded the discussion of our relationship to prior work (Line 59) as suggested by Reviewer c8Cp.

*"… This work is related to randomly localized conformal prediction (RLCP) (Hore & Barber, 2025), where calibration points are chosen according to their proximity in the covariate space. By contrast, MoE-CP identifies relevant calibration points based on proximity in a *learned, label-aware latent regime space* defined by the gating probabilities of the mixture-of-experts. This makes the intervals not only valid, but also sharper, more flexible, and easier to interpret than other local weighted conformal methods; see Section 5 and Remark 5."*

- We added a computational complexity analysis (Line 1227) responding to Reviewer ZRke and Reviewer qsRG.

- We added an analysis of the impact of the randomization step on interval variability and stability (Line 1108), as suggested by Reviewer ZRke.

- We formalized Assumption 1 by introducing a proposition and a detailed proof (Line 766), according to Reviewer c8Cp.

---

### Meta-Review · Area_Chair_JsKX · 2026-01-06

**Summary:**

This paper proposes MoE-CP, a conformal prediction method that uses mixture-of-experts gating probabilities as a similarity measure to construct adaptive, domain-aware prediction intervals with marginal validity guarantees.

Reviewers generally acknowledged the soundness of the approach, the novelty of exploiting MoE gating vectors for conformal weighting, and the strength of the accompanying theoretical analysis.
However, some concerns informed the suggested decision, including questions about whether the contribution is incremental relative to existing localized or weighted conformal methods, the reliance on the assumption that MoE gating meaningfully captures latent domain structure, and the additional computational cost and hyperparameter sensitivity introduced by training MoE models.
Reviewers also raised issues regarding clarity of exposition (particularly the interpretation of the gating mechanism and its distinction from density-based methods), the justification and practical impact of the randomization step, and the limited scope of empirical evaluation on relatively low-dimensional datasets.
While many of these concerns were at least partially addressed in the rebuttal through clarifications, added analysis, and expanded discussion, they reduce enthusiasm for acceptance and suggest that the paper’s strengths lie in solid theoretical grounding and conceptual insight.

**Reviewer Concerns:**

**Concerns addressed**
- Clarification of the MoE formulation and interpretation.
- Distinction from density-based localized conformal methods.
- Justification of the randomization step.
- Hyperparameter choices and computational complexity.

**Concerns still remain**
- Extension to classification settings.
- Perceived incremental contribution.

**Reviewer Scores:**

TKSb: +2.

ZRke: +4. All concerns addressed.

qsRG: +2

c8Cp: may no changes.

---

### Decision · Program_Chairs · 2026-01-26

Accept (Poster)